

# On the genus *Mesopontonia* Bruce, 1967 (Crustacea: Decapoda: Palaemonidae) in Korea, with the description of a new species

Jin-Ho Park[1], Sammy De Grave[2] and Taeseo Park[3]

[1] College of Natural Sciences, Seoul National University, Seoul, Republic of Korea
[2] Oxford University Museum of Natural History, Oxford University, Oxford, United Kingdom
[3] Overseas Biological Resources Team, National Institute of Biological Resources, Incheon, Republic of Korea

## ABSTRACT

*Mesopontonia verrucimanus* and *Mesopontonia kimwoni* sp. nov. are recorded from high-latitude temperate waters in Munseom Islet, Jejudo Island, Republic of Korea, with both species collected on gorgonians and sponges by trimix diving between 50 and 75 m depth. *Mesopontonia kimwoni* sp. nov. is morphologically allied to *M. brevicarpus*, but can be distinguished by the cutting edges of the fingers of the first chela being entire. Significant morphological variation in the rostrum as well as the second pereiopods is documented in *M. verrucimanus*, although cytochrome *c* oxidase subunit I (COI) barcode analysis proves this to be infra-specific variation. A key to species of the genus *Mesopontonia* is provided.

## INTRODUCTION

The deep-sea palaemonid shrimp fauna of the Indo-West Pacific is relatively well documented, with to date 23 genera and about 84 species recorded from depths of more than 100 m by trawling and dredging (*Bruce, 1991*; *De Grave & Fransen, 2011*; *Kou, Li & Bruce, 2016*; *Li, Mitsuhashi & Chan, 2008*; *Marin & Chan, 2014*; *Mitsuhashi & Chan, 2006*; *Okuno, 2017*; *Wang, Chan & Sha, 2015*). Among them is the rarely recorded genus, *Mesopontonia Bruce, 1967* which can be distinguished from related genera by the combination of the absence of both supraorbital and antennal teeth on the carapace, as well as the absence of an exopod on the third maxilliped (*Bruce, 1967*; *Bruce, 1995*; *Chace & Bruce, 1993*).

The most recent classification of carideans by *De Grave & Fransen (2011)* listed six species in the genus, namely *M. gorgoniophila* Bruce, 1967 (type species), *M. gracilicarpus* Bruce, 1990, *M. brucei* Burukovsky, 1991, *M. monodactylus* Bruce, 1991, *M. verrucimanus* Bruce, 1996 and *M. brevicarpus* Li & Bruce, 2006. The type species was originally described from the northern part of the South China Sea (*Bruce, 1967*), in association with two species of the gorgonian genus *Melithaea Milne-Edwards, 1857* (Cnidaria: Octocorallia: Melithaeidae)

Corresponding author
Taeseo Park, polychaeta@gmail.com

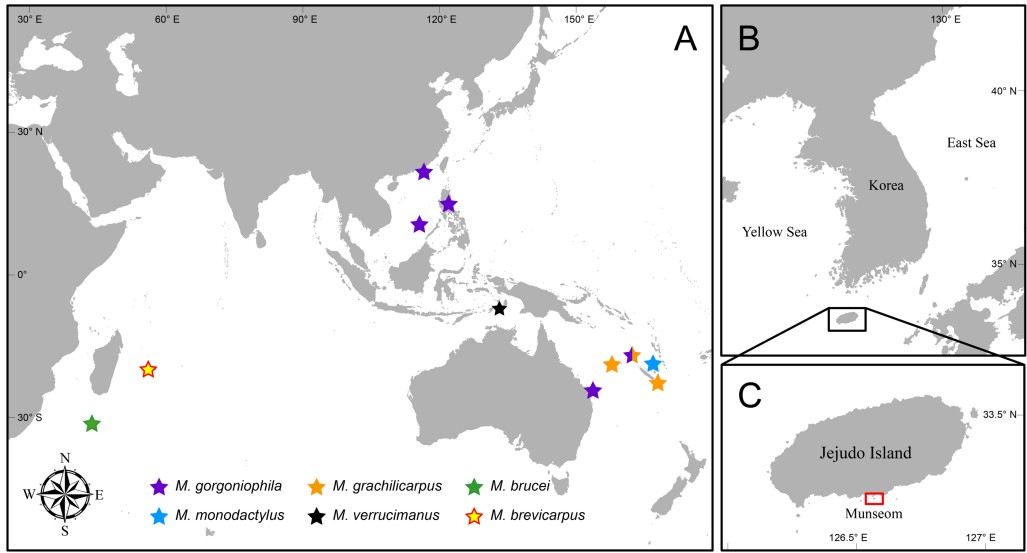

**Figure 1** **Map of Indo West Pacific Ocean (IWP).** Map showing (A) distribution of *Mesopontonia* species, (B) location of Jejudo Island, and (C) the type locality of *M. kimwoni* sp. nov.

in depths of 117–183 m (Fig. 1A). Since then, it has been sparingly reported upon from New Caledonia, the Philippines, and eastern Australia within depths of 130 to 270 m (*Bruce, 1979*; *Bruce, 1984*; *Bruce, 1985*; *Bruce, 1991*). *Mesopontonia gracilicarpus* has been reported from New Caledonia and the Chesterfield Islands in depths of 226–600 m (*Bruce, 1990*; *Bruce, 1991*; *Li & Bruce, 2006*). The southwestern Indian Ocean species, *M. brucei* was recorded from depths of 415–460 m from about 850 km south of Madagascar (*Burukovsky, 1991*). *Mesopontonia monodactylus* was described from the Loyalty Islands (*Bruce, 1991*), associated with species of the hexactinellid sponge genus *Pheronema* Leidy, 1868 (Porifera: Hexactinellida: Pheronematidae) from 460 m (*Bruce, 1990*); *M. verrucimanus* was reported from the Tanimbar Islands, Indonesia in depths of 184–186 m (*Bruce, 1996*). Finally, another western Indian Ocean species, *M. brevicarpus* was recorded from off La Réunion from 270 m (*Li & Bruce, 2006*).

Jejudo Island, the largest island in Korea, is located about 80 km off the southwestern coast of the mainland (Fig. 1B). In summer, the Tsushima current mixes with low-salinity, high-turbidity waters from the Yangtze River to influence the environment around Jejudo Island. In winter, the Yangtze River discharge reduces, resulting in a higher local salinity (*Rebstock & Kang, 2003*; *Lim et al., 2019*). Munseom Islet (Fig. 1C) is located off the south coast of the main island, and consists of volcanic rocks covered with rich invertebrate communities (*Cho et al., 2014*; *Lutaenko, Noseworthy & Choi, 2019*; *Lee et al., 2019*), with a maximum depth of less than 75 m. Thus far, only five symbiotic palaemonid shrimps have been reported from the Jejudo Island area (*Koo & Kim, 2003*; *Lee & Ko, 2011*; *Lee & Ko, 2014*; *Park, De Grave & Kim, 2019a*; *Park, De Grave & Kim, 2019b*), although many more remain unrecorded and will be covered in future contributions.

During a faunal survey for deep-water invertebrate species from previously unexplored habitats around Jejudo Island in 2015–2020, numerous specimens of *Mesopontonia* were collected from gorgonians and sponges by trimix SCUBA diving between 50–75 m depth. Detailed examination of their morphology as well as a phylogenetic analysis including related genera indicated that these belong to *M. verrucimanus* (new record for Korea) and an undescribed species in the same genus, constituting the most northerly record for the genus, as well as the first for temperate waters in the Western Pacific.

In this study, we thus describe both *Mesopontonia* species and present an identification key for the genus *Mesopontonia*. To support systematic studies for deep-sea palaemonid shrimps, molecular analyses and ecological information through direct observations are provided.

## MATERIALS & METHODS

**Sample collection.** Specimens of *Mesopontonia* and other palaemonid shrimps were collected by trimix diving at depths between 50–75 m around Munseom Islet, Jejudo Island, organized by Seoul National University (SNU) and the National Institute of Biological Resources (NIBR) during 2015–2020. Comparative material used in the phylogenetic part was collected by conventional SCUBA diving at depths between 10–30 m in Korea (2018–2019), Palau (2019), Philippines (2018–2019), and Taiwan (2016), organized by Academia Sinica, University of the Philippines Visayas (UPV), Korea Institute of Ocean Science & Technology (KIOST), Marine Biodiversity Institute of Korea (MABIK), and SNU. All specimens were collected together with their host invertebrates and preserved in 80% ethanol. Specimens are deposited in the Marine Arthropod depository Bank of Korea, Seoul National University, Seoul (MADBK), Seoul National University, Seoul (SNU), National Institute of Biological Resources, Incheon (NIBR) and the Zoological Collections of the Oxford University Museum of Natural History, Oxford (OUMNH.ZC).

**Morphological examination.** Shrimps were isolated from the host invertebrate using forceps, with their morphological characteristics observed using stereo microscopes (Leica M205C and M125, Germany) and a light microscope (Olympus BX51, Japan). Digital illustrations were done using a microscope digital camera (Leica MC170, Germany), Helicon focus software (Helicon focus 7.5.6, Ukraine) and drawing tablet (Wacom Intuos Pro PTH-660, China) with Adobe Illustrator software (Adobe Systems, USA), following *Coleman (2006)*.

**Molecular data and phylogenetic analysis.** Molecular phylogenetic analyses were performed to understand the phylogenetic position of the new species, as well as the genus more broadly. Two species of *Mesopontonia* (*M. verrucimanus* and the new species) and 12 deep-sea species from the genera *Altopontonia* Bruce, 1990, *Anchistioides Paul'son, 1875* *Bathymenes* Kou, Li & Bruce, 2016, *Cuapetes Clark, 1919*, *Echinopericlimenes* Marin & Chan, 2014, *Lipkemenes Bruce & Okuno, 2010*, *Palaemonella Dana, 1852*, *Paraclimenes* Bruce, 1995, *Periclimenes Costa, 1844* and *Thaumastocaris Kemp, 1922* were selected as the ingroup. *Stenopus hispidus Olivier, 1811* was used as an outgroup (Table 1). Total genomic DNA was extracted from eggs or pleopod tissue using the QIAamp® DNA

**Table 1   Species used in the phylogenetic analysis with GenBank accession numbers and source.**

| Species | Voucher ID | Location | GenBank accession number | | Sources |
| | | | COI | 16S | |
| --- | --- | --- | --- | --- | --- |
| *Mesopontonia kimwoni* **sp. nov.** (1) | NIBRIV0000862985 | Korea | MT311866 | MT311974 | Present study |
| *M. kimwoni* **sp. nov.** (2) | NIBRIV0000862994 | Korea | MT311865 | MT311973 | Present study |
| *M. verrucimanus* (1) | NIBRIV0000862999 | Korea | MT311867 | MT311975 | Present study |
| *M. verrucimanus* (2) | NIBRIV0000862991 | Korea | MT311868 | MT311976 | Present study |
| *M. verrucimanus* (3) | NIBRIV0000862990 | Korea | MT311869 | MT311977 | Present study |
| *M. verrucimanus* (4) | SNU KR JH1095 | Korea | MT311870 | MT311978 | Present study |
| *M. verrucimanus* (5) | MADBK 120533_007 | Korea | MT311871 | MT311979 | Present study |
| *Paraclimenes* sp. (1) | SNU-KR_JH357 | Korea | MT311872 | MT311980 | Present study |
| *Paraclimenes* sp. (2) | SNU-KR_JH466 | Korea | MT311873 | MT311981 | Present study |
| *Altopontonia disparostris* | RMNH.CRUS.D.51028 | Norfolk Ridge | KM921671 | KU064797 | *Horká et al. (2016)* |
| *Cuapetes tenuipes* | UO V08-48 | Vietnam | KU064965 | KU064814 | *Horká et al. (2016)* |
| *Palaemonella rotumana* | MTQ W-33176 | Australia | KR088755 | KU064830 | *Horká et al. (2016)* |
| *Anchistioides willeyi* | SNU-Pal_Pal12 | Palau | MT311864 | MT311972 | Present study |
| *Lipkemenes lanipes* | SNU-PH_PH99 | Philippines | MT311863 | MT311971 | Present study |
| *Thaumastocaris streptopus* | SNU-TW_TW62 | Taiwan | MT311862 | MK602867 | Present study, *Park, De Grave & Kim (2019b)* |
| *Bathymenes aleator* | UO-948 | Papua New Guinea | MK940921 | MK940946 | *Ďuriš & Šobáňová (2020)* |
| *Echinopericlimenes aurorae* | UO-1057 | Philippines | MK940923 | MK940948 | *Ďuriš & Šobáňová (2020)* |
| *Echinopericlimenes dentidactylus* | UO-1059 | Philippines | MK940930 | MK940954 | *Ďuriš & Šobáňová (2020)* |
| *Echinopericlimenes calcaratus* | UO-960 | Papua New Guinea | MK940925 | MK940949 | *Ďuriš & Šobáňová (2020)* |
| *Periclimenes laccadivensis* | UO-881 | Papua New Guinea | MK940931 | MK940955 | *Ďuriš & Šobáňová (2020)* |
| *Periclimenes uniunguiculatus* | UO-586 | Papua New Guinea | MK940932 | MK940956 | *Ďuriš & Šobáňová (2020)* |
| *Stenopus hispidus* (outgroup) | UO V10-17 | Vietnam | KJ690260 | KU064859 | *Horká et al. (2016)* |

Micro Kit (QIAGEN, Hilden, Germany), following the manufacturer's instructions. Partial sequences of the COI (∼658 bp) and 16S (∼538 bp) markers were amplified via polymerase chain reaction (PCR) with the primers jgHCO2198/jgLCO1490 (*Geller et al., 2013*) and 16S-ar/16S-1472 (*Crandall & Fitzpatrick Jr, 1996*; *Palumbi et al., 2002*), respectively. PCR reactions and sequence data analysis were performed following *Park, De Grave & Kim (2019a)*.

**Zoobank registration.** The electronic version of this article in Portable Document Format (PDF) will represent a published work according to the International Commission on Zoological Nomenclature (ICZN), and hence the new names contained in the electronic version are effectively published under that Code from the electronic edition alone. This published work and the nomenclatural acts it contains have been registered in ZooBank, the online registration system for the ICZN. The ZooBank LSIDs (Life Science Identifiers) can be resolved and the associated information viewed through any standard web browser by appending the LSID to the prefix http://zoobank.org/. The LSID for this publication is: urn:lsid:zoobank.org:pub:3CB43670-472F-49AE-80F2-EAE9597E12BD. The online version of this work is archived and available from the following digital repositories: PeerJ, PubMed Central and CLOCKSS.

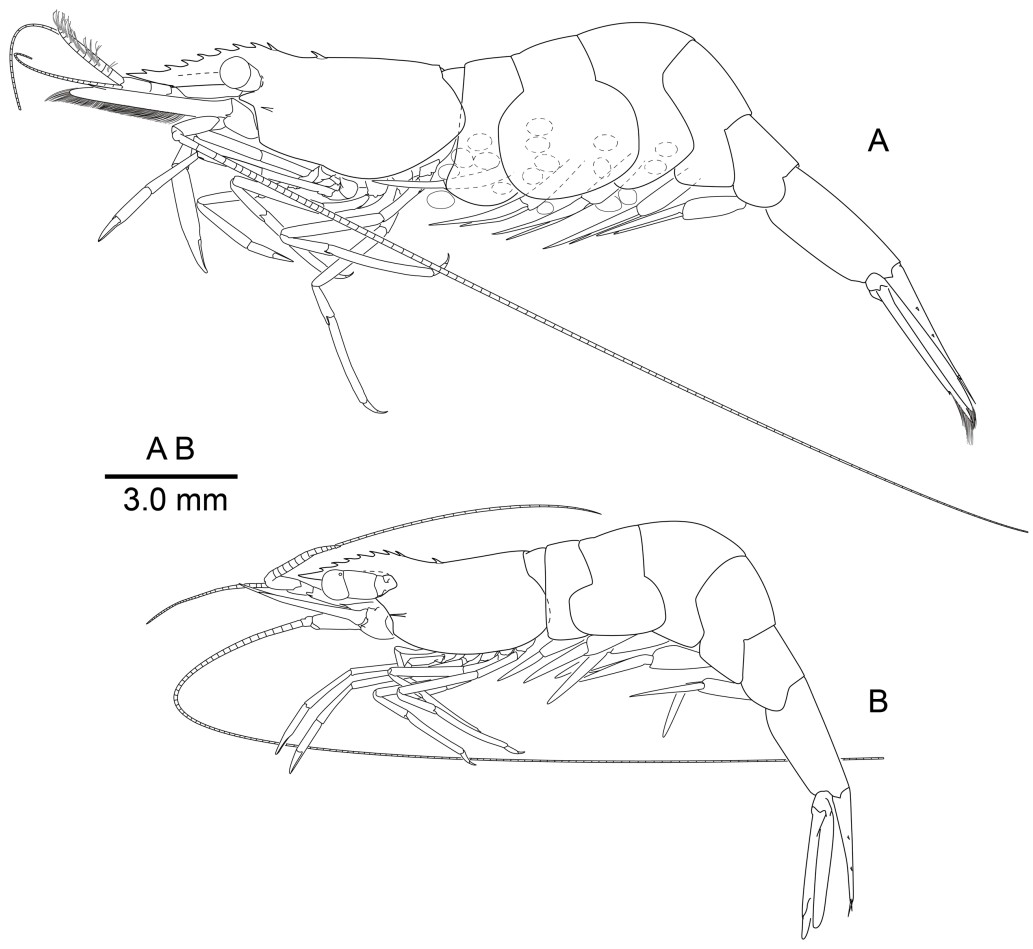

**Figure 2** *Mesopontonia verrucimanus* **Bruce, 1996, habitus, lateral view.** (A) Ovigerous female, pocl 3.7 mm (NIBRIV0000862989); (B) male, pocl 3.2 mm (NIBRIV0000862991).

# RESULTS

## Taxonomy

Family Palaemonidae *Rafinesque, 1815*
Genus *Mesopontonia* Bruce, 1967
***Mesopontonia verrucimanus* Bruce, 1996**
Figs. 2–7

*Mesopontonia verrucimanus* Bruce, 1996: 198, 216–218, figs. 8, 29c (type locality: Tanimbar Islands, Indonesia, 7°59′S, 133°02′E, 184–186 m).

**Material examined.** 2 males, 2 females (pocl 2.0–2.5, R 1+7–8/0); Dec. 20, 2015; Munseom Islet, Jejudo Island, Korea (33°13′36″N 126°34′9″E), 55 m, on *Raspailia* (*Raspaxilla*) *hirsuta* (*Thiele, 1898*), leg. JH Park (MADBK 120533_001); 2 males, 1 female (pocl 2.5, R 1+6–8/0); Apr. 12, 2017; same location (33°13′39″N 126°34′7″E), 58 m, on

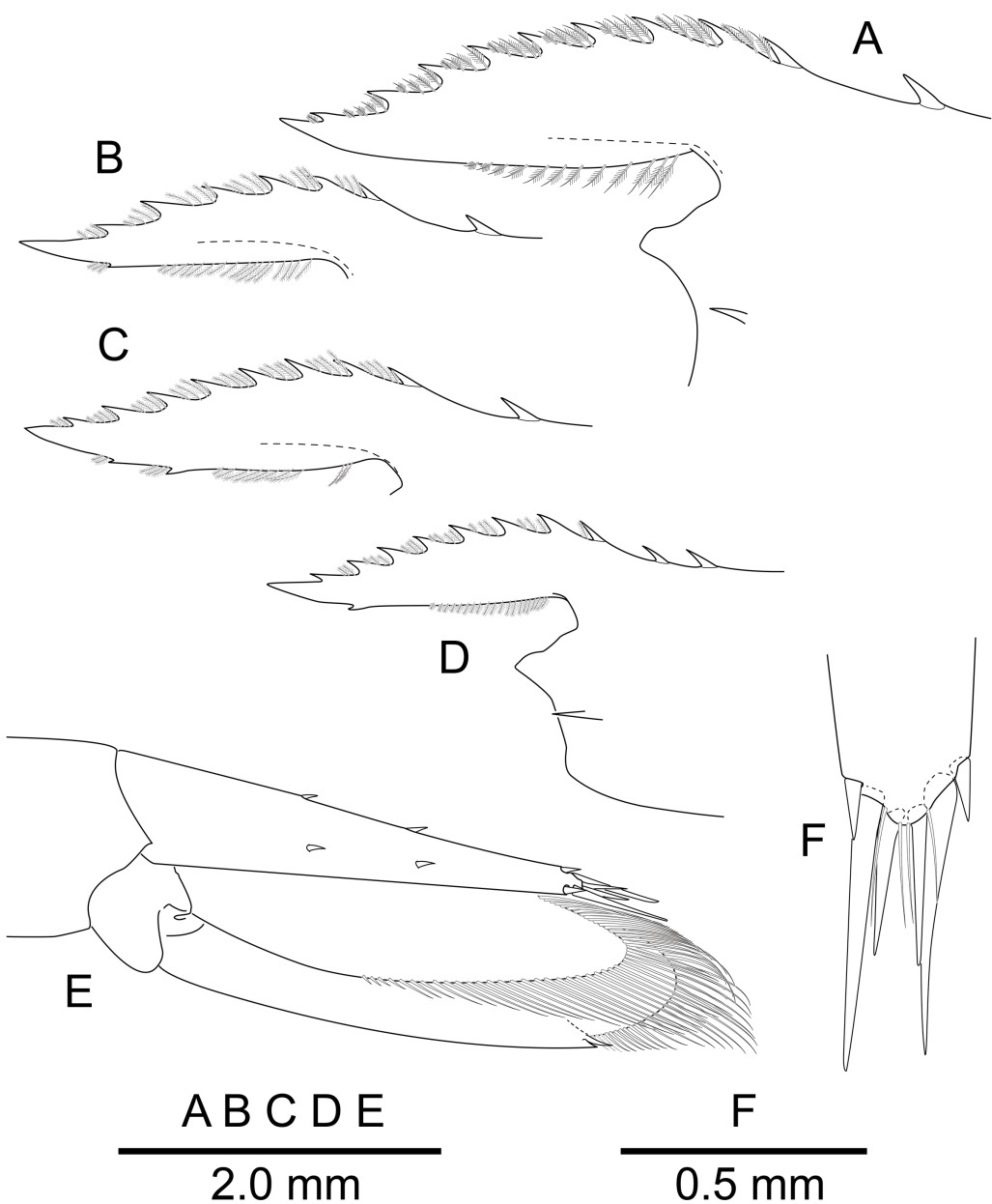

**Figure 3** *Mesopontonia verrucimanus* **Bruce, 1996.** (A) Frontal margin and rostrum, lateral view; (B, C) same, lateral view; (D) same, lateral view; (E) telson and uropods, dorso-lateral view; (F) distal end of telson, dorsal view. (A, E, F) Ovigerous female, pocl 3.8 mm (NIBRIV0000862990); (B) Male, pocl 2.7 mm (NIBRIV0000862982); (C) Female, 2.78 mm (NIBRIV0000862978); (D) Male, pocl 2.5 mm (SNU KR JH1095).

*R. (R.) hirsuta*, leg. JH Park (MADBK 120533_002); 1 male, 2 females (pocl 2.3–2.5, R 1+7–8/0–1); Apr. 12, 2017; same location (33°13′39″N 126°34′7″E), 58 m, on *R. (R.) hirsuta*, leg. JH Park (OUMNH.ZC.2018-03-04-06); 1 male, 4 females (pocl 2.5–3.5, R 1+7–8/0–1); May 30, 2017; same location (33°13′36″N 126°34′9″E), 54 m, on *Ellisella* cf. *limbaughi* Bayer & Deichmann, 1960, leg. JH Park (MADBK 120533_003); 1 male, 1

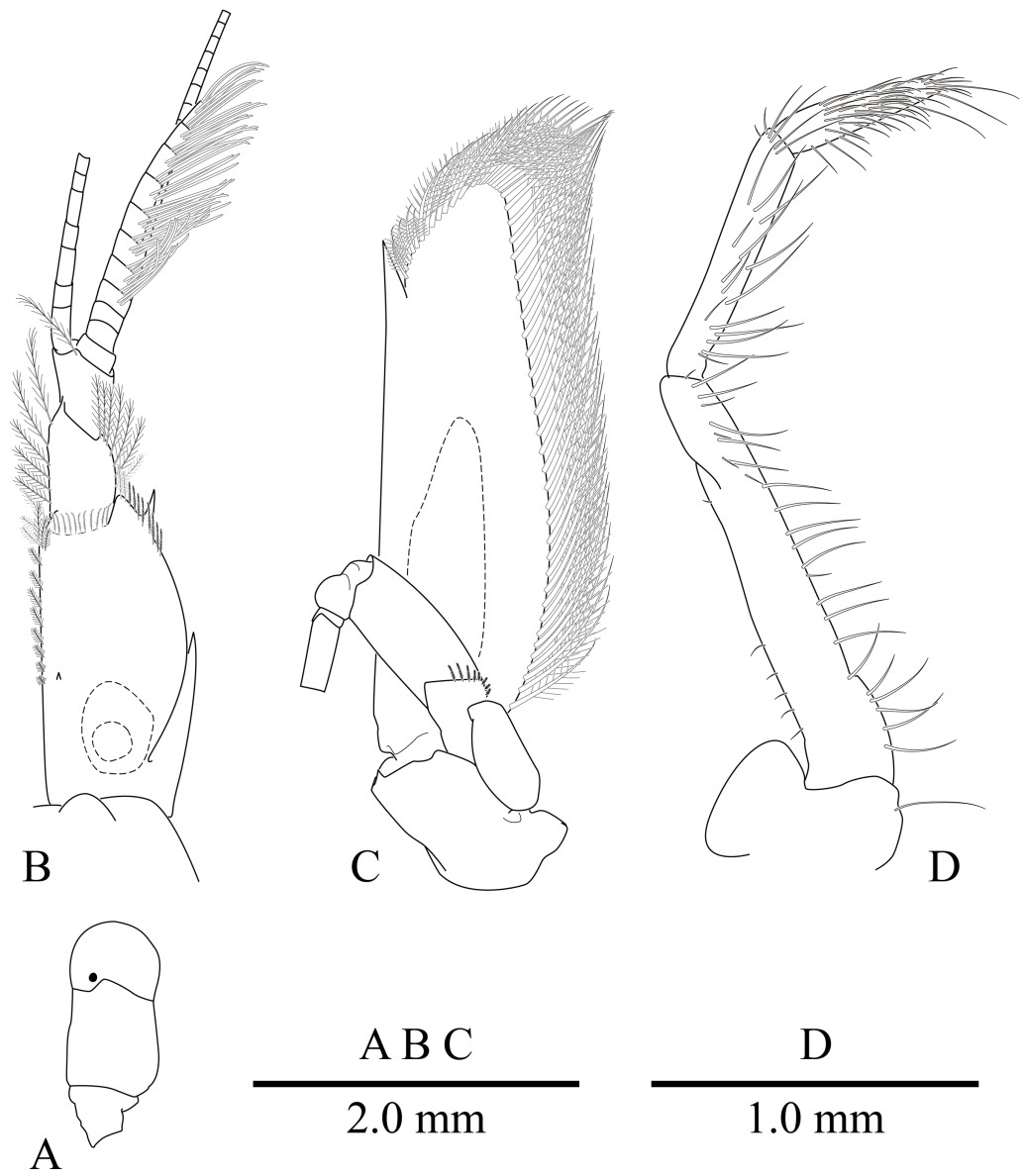

**Figure 4** *Mesopontonia verrucimanus* **Bruce, 1996.** (A) Left eye, dorsal view; (B) left antennule, dorsal view; (C) left antenna, dorsal view; (D) left third maxilliped. (A–C) Ovigerous female, pocl 3.8 mm (NIB-RIV0000862990); (D) Ovigerous female, pocl 3.7 mm (NIBRIV0000862989).

female (pocl 2.5, R 1+7–8/1); Jan. 18, 2018; same location (33°13′36″N 126°34′9″E), 60 m, on *E.* cf. *limbaughi*, leg. JH Park (MADBK 120533_004); 2 males, 3 females (pocl 2.0–2.5, R 1+7–8/0–1); Jan. 18, 2018; same location (33°13′36″N 126°34′9″E), 60 m, on *E.* cf. *limbaughi*, leg. JH Park (NIBRIV0000837760–0000837764); 1 female (pocl 3.0, R 1+8/1); Mar. 31, 2018; same location (33°13′36″N 126°34′9″E), 60 m, on *E.* cf. *limbaughi*, leg. JH Park (MADBK 120533_005); 1 male (pocl 2.6, R 1+8/0); Jun. 20, 2018; same location (33°13′31″N 126°34′11″E), 60 m, on *E.* cf. *limbaughi*, leg. JH

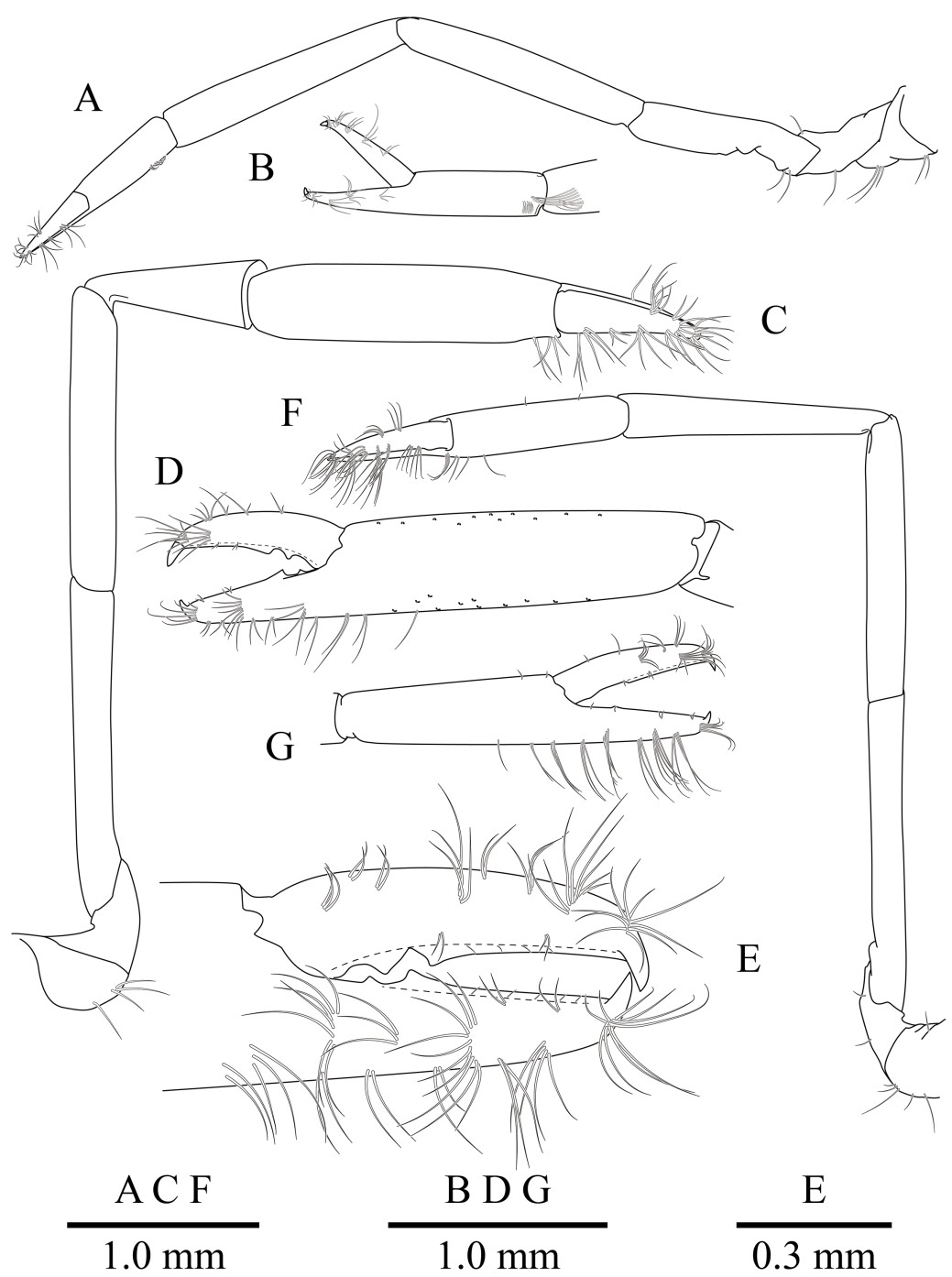

**Figure 5** *Mesopontonia verrucimanus* **Bruce, 1996, ovigerous female, pocl 3.7 mm (NIBRIV0000862989).** (A) Left first pereiopod. (B) same, chela; (C) major right second pereiopod; (D) same, chela and carpus; (E) same, fingers; (F) minor left second pereiopod; (G) same, chela.

Park (MADBK 120533_006); 1 male, 2 females (pocl 2.78–2.87, R 1+7–8/0–2); Jun. 20, 2018; same location (33°13′31″N 126°34′11″E), 60 m, on *E.* cf. *limbaughi*, leg. JH Park

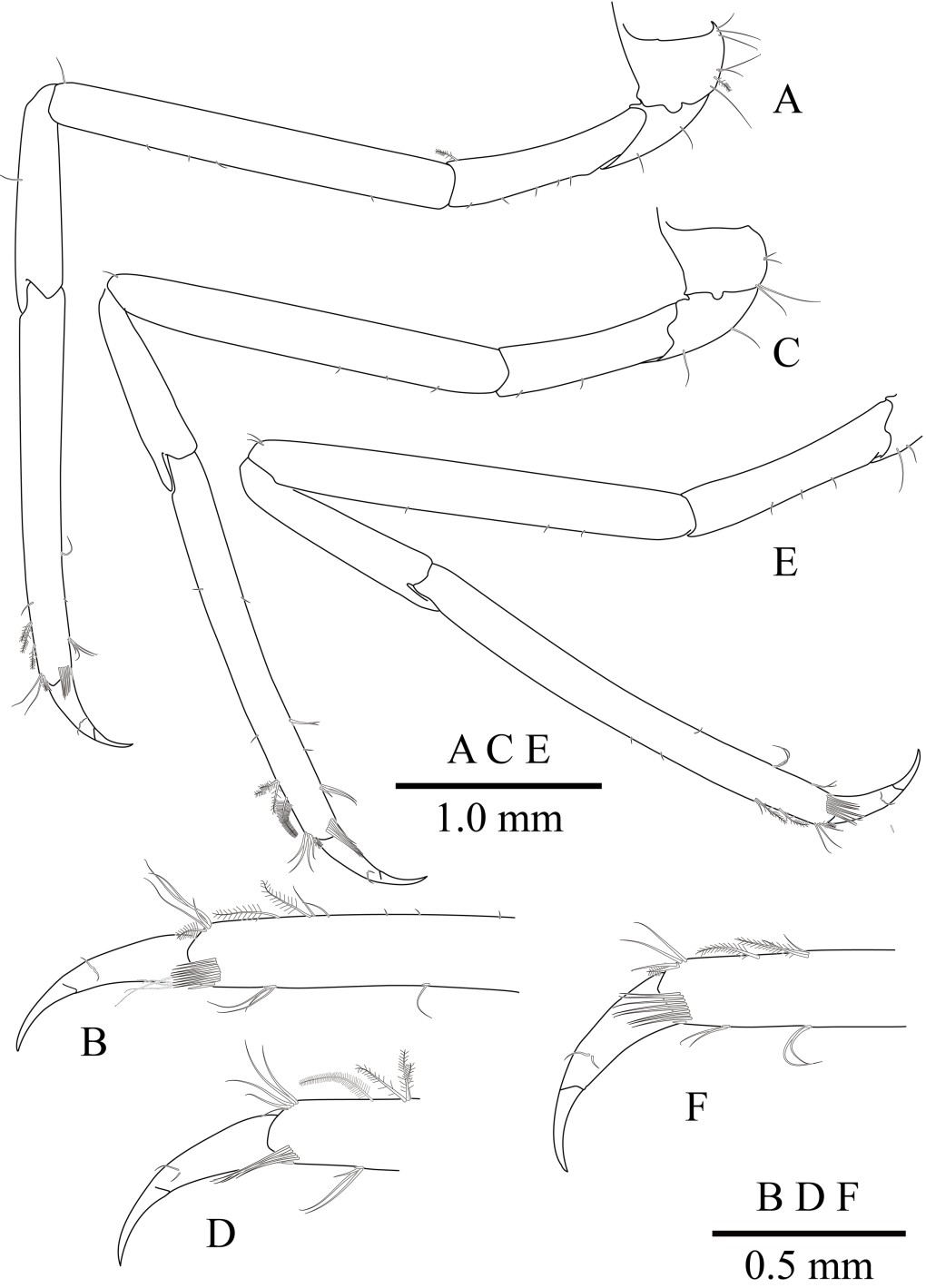

**Figure 6** *Mesopontonia verrucimanus* **Bruce, 1996, ovigerous female, pocl 3.7 mm (NIBRIV0000862989).** (A) Left third pereiopod; (B) same, dactylus and distal propodus; (C) left fourth pereiopod; (D) same, dactylus and distal propodus; (E) left fifth pereiopod; (F) same, dactylus and distal propodus.

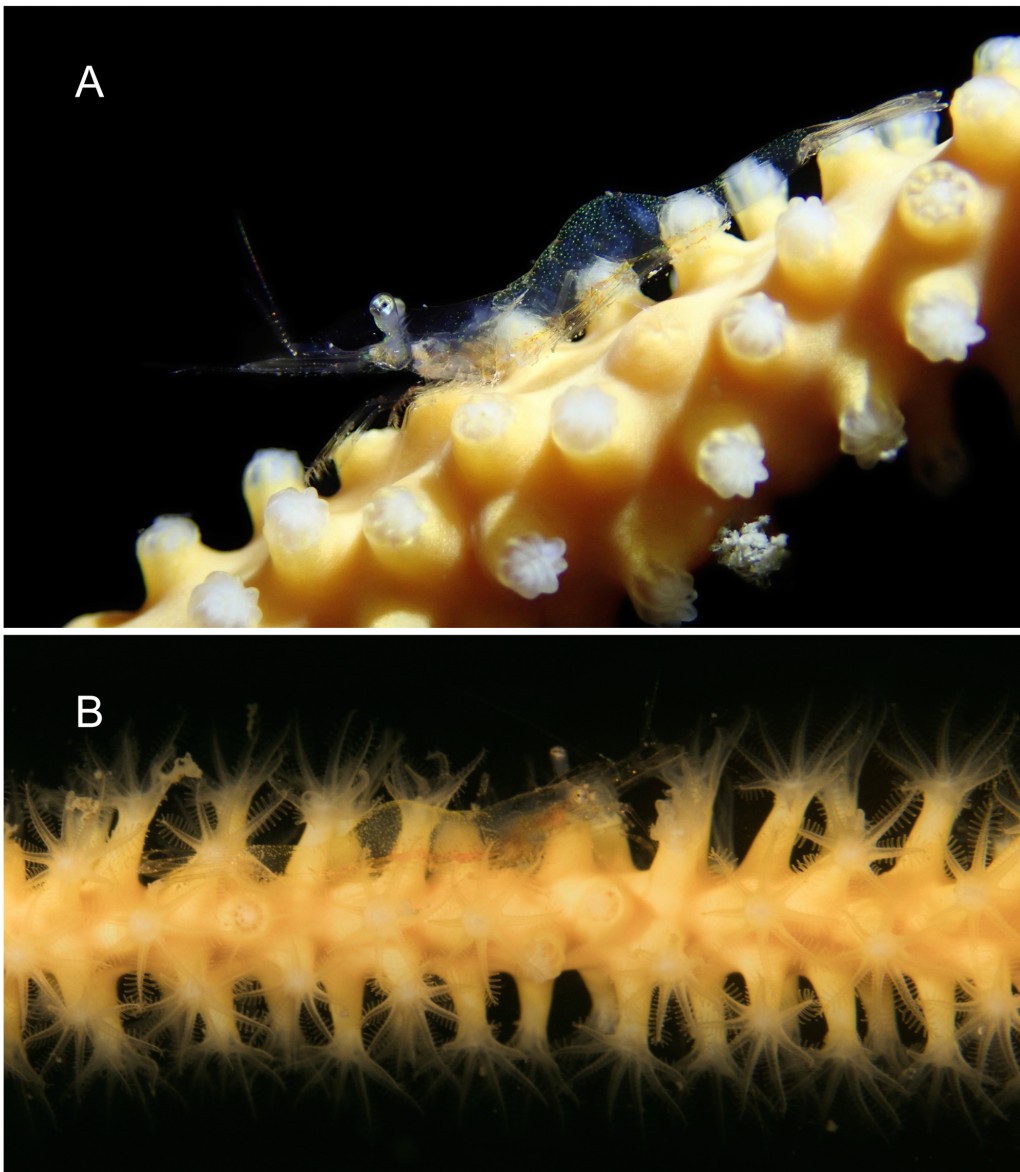

**Figure 7** *Mesopontonia verrucimanus* **Bruce, 1996 on** *Ellisella limbaughi* **Bayer & Deichmann, 1960 from Munseom Islet, Jejudo Island.** (A) Female specimen, pocl 2.5 mm (MADBK 120533_004); (B) male specimens, pocl 3.0 mm (MADBK 120533_003). Photographic Credits: (A) Jin-Ho Park, (B) Jong Moon Choi.

(NIBRIV0000837778–NIBRIV0000837780); 1 male, 1 female, 1 ovigerous female (pocl 2.67–3.47, R 1+6–8/*0x1*); Jun. 21, 2018; same location (33°13′31″N 126°34′11″E), 55 m, on *E.* cf. *limbaughi*, leg. JH Park (NIBRIV0000862982–NIBRIV0000837784); 1 male (pocl 3.13, R 1+7/0); Sep. 12, 2018; same location (33°13′29″N 126°33′52″E), 60 m, on *Myriopathes lata* (*Silberfeld, 1909*), leg. JH Park (NIBRIV0000862987); 1 male (pocl 3.41, R 1+7/0); Sep. 13, 2018; same location (33°13′29″N 126°33′52″E), 60 m, on *E.* cf. *limbaughi*, leg.

JH Park (NIBRIV0000862988); 2 ovigerous females (pocl 3.7–3.8, R 1+8–9/0); Sep. 13, 2018; same location (33°13′29″N 126°33′52″E), 60 m, on *Cirrhipathes* cf. *anguina* (*Dana, 1846*), leg. JH Park (NIBRIV0000862989–0000837790); 2 males (pocl 3.2, R 1+8–9/0); Jul. 26, 2019; same location (33°13′35″N 126°34′11″E), 60 m, on *E.* cf. *limbaughi*, leg. JH Park (NIBRIV0000862991–0000837792); 1 ovigerous female (pocl 3.4, R 1+9/1); Jul. 27, 2019; same location (33°13′35″N 126°34′11″E), 55 m, on *E.* cf. *limbaughi*, leg. JH Park (NIBRIV0000862993); 1 female (pocl 1.5, R 1+7/0); Aug. 16, 2019; same location (33°13′41″N 126°34′6″E), 53 m, on *C. anguina*, leg. JH Park (NIBRIV0000862995); 1 male, 3 females, 2 ovigerous females (pocl 1.5–3.7, R 1+8–9/0–1); Oct. 22, 2019; same location (33°13′37″N 126°34′11″E), 67 m, on *E.* cf. *limbaughi*, leg. JH Park (NIBRIV0000862996–NIBRIV0000863001); 2 males, 1 female (pocl 2.4–3.7, R 1+8–9/0–2); Jan. 15, 2020; same location (33°13′34″N 126°33′45″E), 75 m, on *E.* cf. *limbaughi*, leg. JH Park (MADBK 120533_007); 3 females (pocl 2.5–2.8, R 1+7–8/0); Jan. 15, 2020; same location (33°13′34″N 126°33′45″E), 75 m, on *C. anguina*, leg. JH Park (SNU KR JH1100–1102); 1 male (pocl 2.5, R 2–8/1); Jan. 15, 2020; same location (33°13′34″N 126°33′45″E), 75 m, on *Raspailia* sp., leg. JH Park (SNU KR JH1095); 1 female (pocl 2.3, R 1–8/0); Jan. 15, 2020; same location (33°13′34″N 126°33′45″E), 75 m, on *Raspailia* sp., leg. JH Park (SNU KR JH1096); 1 female (pocl 2.9, R 1–9/0); Jan. 15, 2020; same location (33°13′34″N 126°33′45″E), 75 m, on *C. anguina*, leg. JH Park (SNU KR JH1104).

**Description of Korean specimens.** Body (Fig. 2) small-sized, subcylindrical form. Rostrum (Figs. 2 and 3A–3D) straight, horizontal, almost as long as pocl, reaching or overreaching distal end of antennular peduncle, 6–9 dorsal teeth, spaced along entire length, 0–2 ventral teeth.

Carapace (Figs. 2 and 3A–3D) smooth, glabrous, with epigastric tooth at anterior 0.3 of pocl; without supraorbital and antennal teeth; inferior orbital angle produced; hepatic tooth large, acute, extending to anterior margin of carapace; pterygostomial angle bluntly rounded.

Abdomen (Fig. 2) smooth; pleura of first five segments rounded; sixth pleura with pointed posterolateral angle, posteroventral angle subacute.

Telson (Figs. 2 and 3E, 3F) about 0.75 of pocl, 4.0 times as long as proximal width; two pairs of small dorsal spiniform setae at 0.4 and 0.65 of telson length respectively, with three pairs of posterior spiniform setae, lateral pair shortest, medial pair long and stout.

Eye (Figs. 2 and 4A) with hemispherical cornea, dorsolaterally with nebenauge, diameter about 0.20 of pocl.

Antennule (Figs. 2 and 4B) with proximal peduncle bearing distolateral tooth, with small acute tooth at ventromedial margin; stylocerite narrow, bearing sharp point, reaching to 0.45 times of proximal segment; intermediate segment short, 0.4 times of proximal segment length, 0.8 of distal segment; upper flagellum biramous, proximal five segments fused, lower flagellum slender, filiform.

Antenna (Figs. 2 and 4C) basicerite with sharp pointed distodorsal margin; ischiocerite and merocerite unarmed; carpocerite reaching to about 0.4 of scaphocerite length; scaphocerite 4 times as long as maximal width, lateral margin rounded, medial margin convex, distolateral tooth large, at 0.9 of lamella length.

Mouthparts typical for genus. Third maxilliped (Fig. 4D) without exopod, reaching to middle of carpocerite; ultimate segment about 0.4 of antepenultimate segment length, tapering distally, with transverse rows of setae; penultimate segment about 0.6 of antepenultimate segment length, with sparsely row of long setae ventromedially; ischiomerus completely fused to basis, antepenultimate segment feebly compressed distally, with long setae ventromedially; coxa with rounded medial lobe, with rounded lateral plate.

First pereiopod (Figs. 2, 5A and 5B) reaching to distal end of scaphocerite; fingers about 0.81 of palm length, tips hooked, cutting edge entire, with transverse row of setae and group of terminal setae; palm ventrolaterally with transverse row of serrulate setae; carpus 1.3 times length of chela with row of serrulate setae along distomesial margin; merus as long as carpus; ischium about 0.7 times length of merus; basis and coxa without special features.

Second pereiopods (Figs. 2A, 5C–5G) well developed, dissimilar in shape, unequal in size. Major second pereiopod (Figs. 5C–5E) overreaching distal end of rostrum by middle of propodus; chela about 0.63 of pocl, with group of terminal setae; fingers about 0.5 of palm length; dactylus slender, about 3.6 times longer than proximal depth, distally curved with acute tip, proximally with two acute teeth at 0.3 and 0.4, distally entire without dorsolateral flange; fixed finger with acute tip, proximally with two small teeth at 0.2 and 0.3, distally entire; palm subcylindrical, about 4.0 times longer than distal width, covered with minutely blunt tubercles and simple setae; carpus about 0.45 of palm length, about 2 times longer than distal width; merus about 2.0 times as long as carpus, as long as palm length, about 6.2 times longer than distal width; ischium as long as carpus length, about 6.0 times longer than distal width; basis and coxa without special features.

Minor second pereiopod (Figs. 5F, 5G) slightly overreaching distal end of scaphocerite; chela about 0.4 of pocl, 0.7 of major chela length, with group of terminal setae; fingers about 0.7 of palm length, distally curved with acute tips, cutting edge entire; palm subcylindrical, about 3 times longer than distal width, smooth slightly tapering proximally; carpus about 1.4 of palm length, about 0.9 of chela length, about 5.6 times longer than distal width; merus about 1.1 times as long as carpus length, about 7.8 times longer than distal width; ischium about 1.1 times as long as merus length, about 7.8 times longer than distal width; basis and coxa without special features.

Ambulatory pereiopods (Figs. 2 and 6) subequal in shape and size. Third pereiopod (Figs. 6A, 6B) overreaching distal end of rostrum by distal end of propodus; dactylus about 0.26 of propodus length, about 4.2 times longer than proximal width, not biunguiculate; propodus about 10 times longer than distal width, ventral border with four distolateral spiniform setae including pair distoventral one, with long setae distally, with distodorsal plumose setae and distal setae; carpus about 0.54 times length of propodus, unarmed; merus as long as carpus length, unarmed; ischium about 0.5 length of propodus, unarmed; basis and coxa without special features. Fourth and fifth pereiopods (Figs. 6C–6F) similar to third pereiopod.

Uropod (Figs. 2 and 3E) overreaching distal end of telson; exopod with distolateral tooth and movable acute tooth.

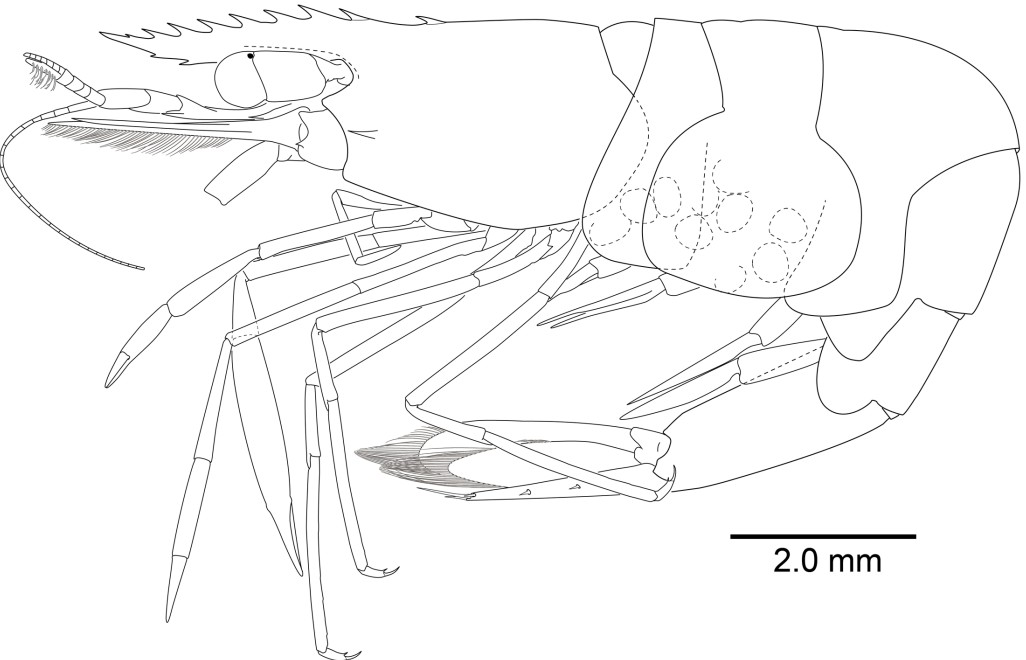

**Figure 8** *Mesopontonia kimwoni* **sp. nov., habitus, lateral view.** Holotype, ovigerous female, pocl 2.5 mm (NIBRIV0000862985).

**Variation.** *Bruce (1996)* described *Mesopontonia verrucimanus* based on a single specimen (holotype), with nine dorsal teeth, unarmed ventrally and markedly unequal, dissimilar second pereiopods. The Korean specimens exhibit morphological variation in rostral dentition and the major second pereiopod. The number of dorsal and ventral rostral teeth (Figs. 2 and 3A–3D) varies from 6–9 and 0–2 respectively. A single specimen (Fig. 3D) harbours two epigastric teeth on the carapace. Several specimens (Fig. 2B) bear two symmetrical second pereiopods, which are very similar to the minor second pereiopods in the original description and in other Korean specimens (Fig. 2A).

**Color in life.** Whole body and appendages almost transparent with scattered emerald green chromatophores (Fig. 7); longitudinal pale red band along the ventral surface of the body from the carapace to the fifth abdominal somite.

**Geographical distribution.** Presently known from the Tanimbar Islands (Indonesia) and Jejudo Island (Republic of Korea) (Fig. 1).

**Habitat and host.** The present specimens were obtained from gorgonian and sponge colonies below 50 m (Figs. 14A, 14B), with the deepest samples from 75 m depth. The present specimens demonstrate a lack of host specificity and the species cannot be considered as restricted to gorgonians, as previously postulated. Most specimens were collected on the orange colored sea whip, *Ellisella* cf. *limbaughi* (*Bayer & Deichmann, 1960*), with further specimens obtained on the white colored gorgonian, *Cirrhipathes* cf. *anguina*, as well as the orange colored sponge, *Raspailia* (*Raspaxilla*) *hirsuta Thiele, 1898*. A single specimen was collected on the white colored antipatharian, *Myriopathes lata*.

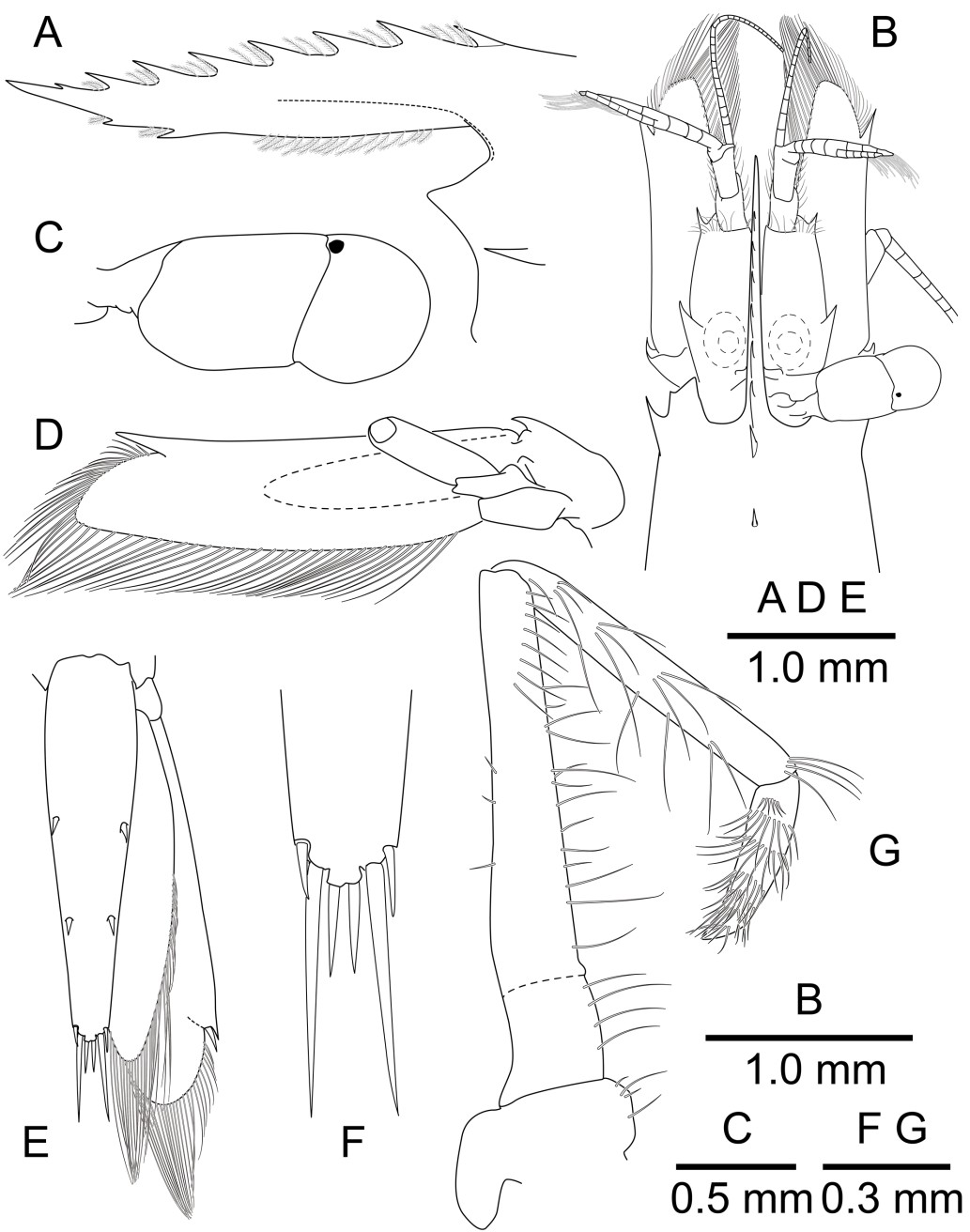

**Figure 9** *Mesopontonia kimwoni* sp. nov., ovigerous female pocl 2.5 mm (NIBRIV0000862985), holotype. (A) frontal margin and rostrum, lateral view; (B) frontal region, dorsal view; (C) left eye, dorsal view; (D) left antenna, ventral view; (E) telson and uropod, dorsal view; (F) distal end of telson, dorsal view; (G) left third maxilliped, mesial view.

**Remarks**. *Mesopontonia verrucimanus* can be immediately separated from most other species in the genus which have a biunguiculate dactylus of ambulatory pereiopods, except *M. monodactylus* with which it shares a non-biunguiculate dactylus. *M. monodactylus*

differs from *M. verrucimanus* primarily by having a distinct dorsolateral flange on the chela of the major second pereiopod (*Bruce, 1991*; *Bruce, 1996*).

### *Mesopontonia kimwoni* sp. nov.
urn:lsid:zoobank.org:act:BBA317A3-7140-4D97-BCF4-DB6EEF6617F5
Figs. 8–12

## Material examined
**Type material.** *Holotype.* 1 ovigerous female (pocl 2.5, R 1+8/2); Jun. 21, 2018; Munseom Islet, Jejudo Island (33°13′31″N 126°34′11″E), 55 m, on *Myriopathes lata*, leg. JH Park (NIBRIV0000862985). *Paratype.* 1 ovigerous female (pocl 2.8, R 1+8/2); Aug. 16, 2019; same location (33°13′41″N 126°34′6″E), 53 m, on *M. lata*, leg. JH Park (NIBRIV0000862994).

**Description.** Body (Fig. 8) small-sized, subcylindrical form. Rostrum (Figs. 8 and 9A) straight, horizontal, almost as long as pocl, reaching or slightly beyond end of antennular peduncle, 8 dorsal teeth, along entire length, 2 subterminal ventral teeth.

Carapace (Fig. 8) smooth, glabrous, with epigastric tooth at anterior 0.3 of pocl; without supraorbital and antennal teeth; inferior orbital angle produced; hepatic tooth large, acute, extending to anterior margin of carapace; pterygostomial angle bluntly rounded.

Abdomen (Fig. 8) smooth; pleura of first five segments rounded; sixth pleura with pointed posterolateral angle, posteroventral angle subacute.

Telson (Figs. 8 and 9D) 0.8 of pocl, 4 times as long as proximal width; two pairs of small dorsal spiniform setae at 0.45 and 0.7 of telson length respectively, with three pairs of posterior spiniform setae, lateral pair shortest, medial pair long and stout.

Eye (Figs. 8 and 9B–9C) with hemispherical cornea, dorsolaterally with nebenauge, diameter about 0.23 of pocl.

Antennule (Fig. 9B) with proximal peduncle bearing distolateral tooth, with small acute tooth at ventromedial margin; stylocerite narrow, bearing sharp point, reaching to 0.45 times of proximal segment; intermediate segment short, 0.3 times of proximal segment length, subequal to distal segment; upper flagellum biramous, proximal four segments fused, lower flagellum slender, filiform.

Antenna (Figs. 8 and 9B) basicerite with sharp pointed distodorsal margin; ischiocerite and merocerite unarmed; carpocerite reaching to about 0.4 of scaphocerite length; scaphocerite 4 times as long as maximal width, lateral margin rounded, medial margin convex, distolateral tooth large, at 0.9 of lamella length.

Mouthparts not dissected. Third maxilliped (Fig. 9F) without exopod, reaching to 0.7 of carpocerite; ultimate segment about 0.35 of antepenultimate segment length, tapering distally, with transverse rows of setae; penultimate segment about 0.7 of antepenultimate segment length, with sparsely row of long setae ventromedially; ischiomerus completely fused to basis, antepenultimate segment feebly compressed distally, with long setae ventromedially; coxa with rounded medial lobe, with rounded lateral plate.

First pereiopod (Figs. 8 and 9A, 9B) overreaching distal end of scaphocerite; fingers about 0.6 of palm length, tips hooked, cutting edge entire, with transverse row of setae and group of terminal setae; palm ventrolaterally with transverse row of serrulate setae; carpus 1.1 times length of chela with row of serrulate setae along distomesial margin; merus 1.1 times length of carpus; ischium about 0.5 times length of merus; basis and coxa without special features.

Second pereiopods (Figs. 8 and 10C–10G) well developed, similar in shape, unequal in size. Major second pereiopod (Figs. 10C–10E) overreaching distal end of rostrum by middle of propodus; chela about 1.3 times as long as pocl, with group of terminal setae; fingers about 0.4 of palm length; dactylus slender, about 3.8 times longer than proximal depth, distally curved with acute tip, proximally with two blunt teeth at proximal 0.3 and 0.4, distally entire without dorsolateral flange; fixed finger with distally curved with acute tip, proximally with single acute tooth at 0.4, distally entire; palm subcylindrical, about 4.2 times longer than distal width, covered with minutely blunt tubercles and short simple setae; carpus about 0.4 of palm length, about 2.8 times longer than distal width; merus about 2.1 times as long as carpus, about 0.8 of palm length, 7.0 times longer than distal width; ischium subequal to carpus length, about 7.0 times longer than distal width; basis and coxa without special features.

Minor second pereiopod (Figs. 10F, 10G) overreaching distal end of rostrum by end of carpus; chela about 0.7 of pocl, 0.7 of major chela length, with group of terminal setae; fingers about 0.7 of palm length, with distally curved with acute tips, cutting edge entire; palm subcylindrical, about 3.75 times longer than distal width, smooth, slightly tapering proximally; carpus about 1.3 of palm length, about 0.75 of chela length, about 7.2 times longer than distal width; merus about 1.1 times as long as carpus length, about 9 times longer than distal width; ischium about 0.9 of merus length, about 10 times longer than distal width; basis and coxa without special features.

Ambulatory pereiopods (Figs. 8 and 11) subequal in shape and size, only third pereiopod with distodorsal plumose setae and distal serrulate setae on propodus. Third pereiopod (Figs. 11A, 11B) overreaching distal end of rostrum by distal half of propodus; dactylus about 0.2 of propodus, about 4 times longer than proximal width, about 0.65 of corpus length, biunguiculate; propodus about 0.8 of pocl, 16 times longer than distal width, ventral border with four distolateral spiniform setae including single distoventral one, with long setae distally; carpus about 0.4 times length of propodus, unarmed; merus about 0.9 times length of propodus, unarmed; ischium about 0.5 length of propodus, unarmed; basis and coxa without special features.

Fourth pereiopod (Figs. 11C, 11D) with dactylus about 0.2 times length of propodus, about 4 times longer than proximal width, about 0.65 of corpus length, biunguiculate; propodus with four distolateral spiniform setae including single distoventral one, with long simple setae distally; carpus about 0.45 times length of propodus, unarmed; merus subequal to propodus length, unarmed; ischium about 0.46 length of propodus; basis and coxa without special feature. Fifth pereiopod (Fig. 11E) similar to fourth pereiopod.

Uropod (Fig. 11D) overreaching distal end of telson; exopod with distolateral tooth and movable acute tooth.

 

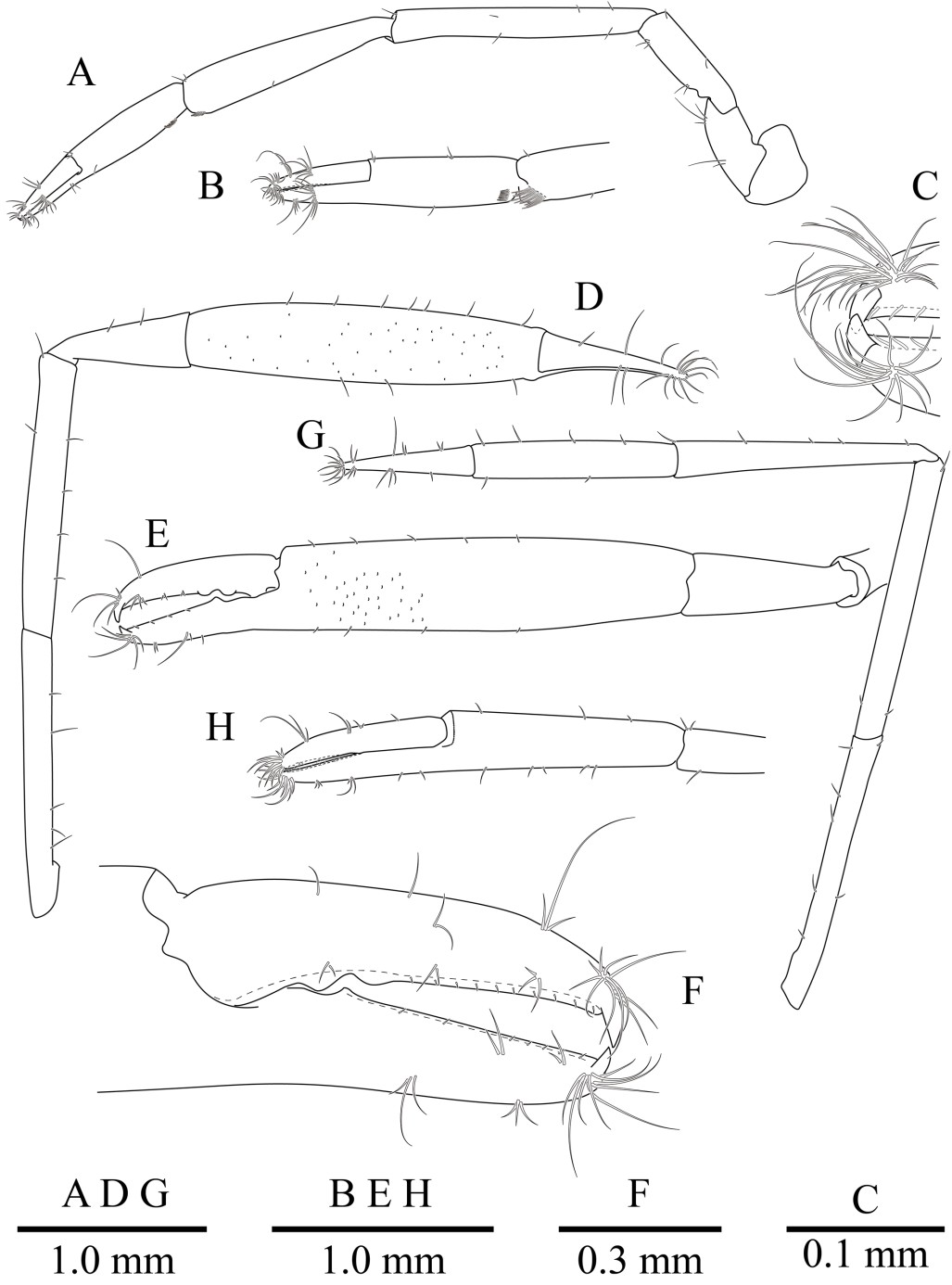

A D G — 1.0 mm
B E H — 1.0 mm
F — 0.3 mm
C — 0.1 mm

**Figure 10** *Mesopontonia kimwoni* sp. nov., ovigerous female pocl 2.5 mm (NIBRIV0000862985), holotype. (A) left first pereiopod; (B) same, chela; (C) major right second pereiopod; (D) same, chela and carpus; (E) same, fingers; (F) minor left second pereiopod; (G) same, chela.

**Etymology.** The specific name "kimwoni" is in honor of Dr. Kim, Won, professor in the School of Biological Sciences, Seoul National University. He made a considerable contribution to the systematics of Korean crustaceans.

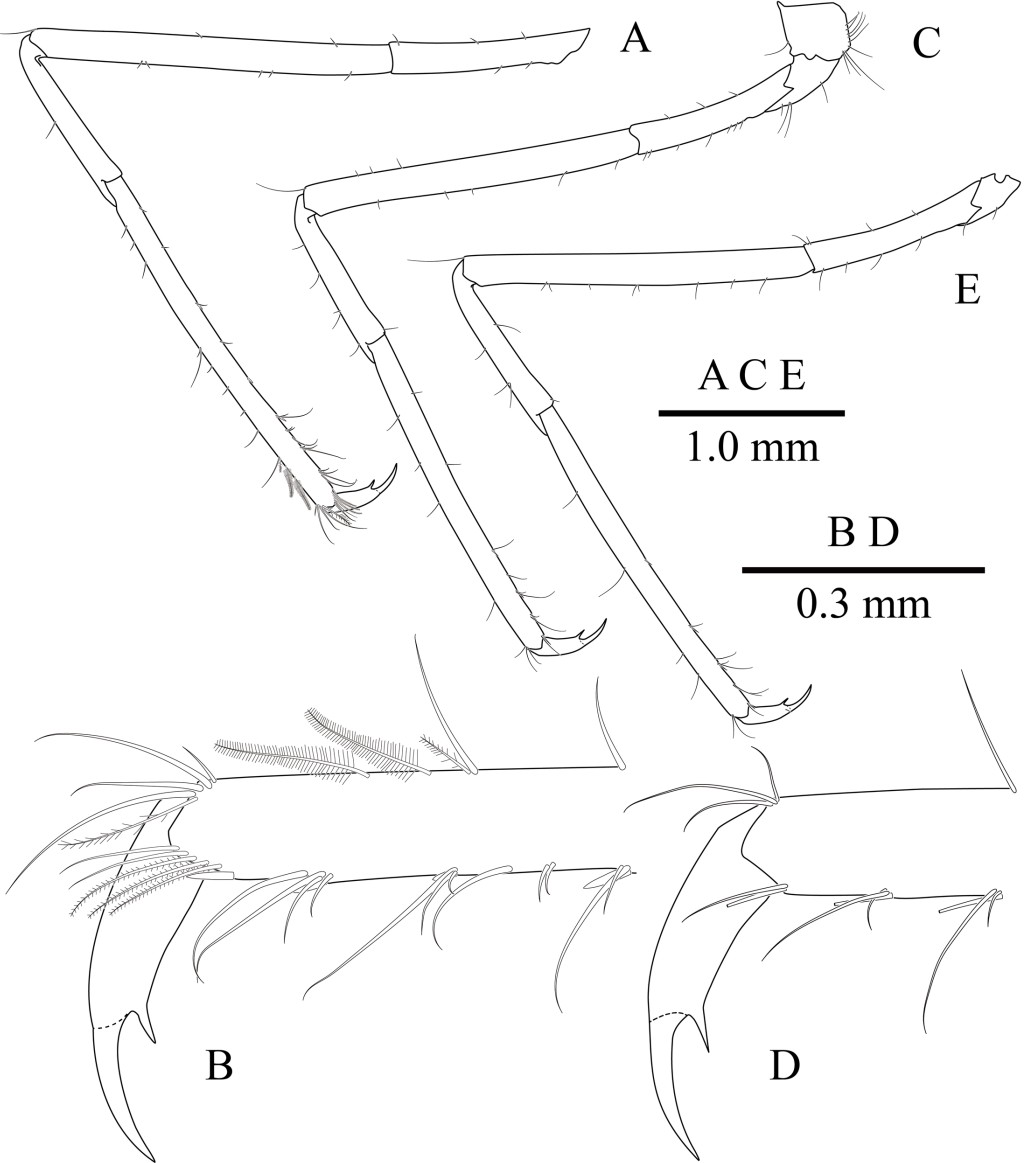

**Figure 11** *Mesopontonia kimwoni* **sp. nov., ovigerous female pocl 2.5 mm (NIBRIV0000862985), holotype.** (A) left third pereiopod; (B) same, dactylus and distal propodus; (C) left fourth pereiopod; (D) same, dactylus and distal propodus; (E) left fifth pereiopod.

**Color in life.** Whole body and appendages almost transparent (Figs. 12 and 13); longitudinal red bands along the ventral surface of the body from antennular peduncle to the fifth abdominal somite; tiny white and red chromatophore scattered along the dorsal surface of cornea of eyes, proximal segment of antennular peduncle and abdomen.

**Type locality.** Munseom Islet, Jejudo Island, Korea

**Geographical distribution.** Presently only known for the type locality.

**Habitat.** The two specimens of *M. kimwoni* sp. nov. were collected from the gorgonian antipatharian, *Myriopathes lata* in 53–55 m (Fig. 14C).

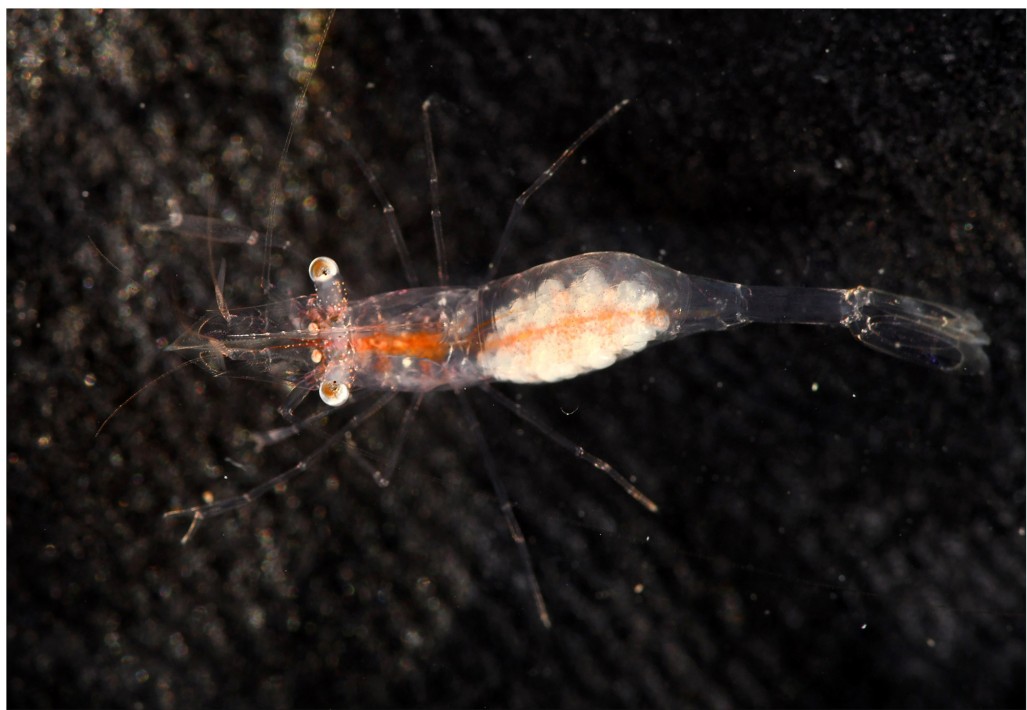

**Figure 12** *Mesopontonia kimwoni* **sp. nov. from Munseom Islet, Jejudo Island.** Holotype, ovigerous female, pocl 2.5 mm (NIBRIV0000862985). Photographic credit: Jin-Ho Park.

**Remarks.** Based on the presence of the biunguiculate dactyli of ambulatory pereiopods, the new species is morphologically allied to four species: *M. brevicarpus*, *M. brucei*, *M. gorgoniophila*, and *M. gracilicarpus* Bruce, 1990; and can be easily separated from the remaining two species in the genus (*M. monodactylus*, *M. verrucimanus* which have simple dactyli. The new species differs from *M. gorgoniophila* by the straight distodorsal carina in the major second pereiopod (vs. oblique distodorsal carina in *M. gorgoniophila*), with two teeth on the cutting edge of dacytlus of major second pereiopod (vs. with single tooth in *M. gorgoniophila*), dorsolateral dactylar flange of major second pereiopod absent (vs. present in *M. gorgoniophila*), as well as the proportions of the carpus of the minor second pereiopod (about 1.3 time palm length in *M. kimwoni* **sp. nov.** vs. about 0.8 in *M. gorgoniophila*). The new species differs from *M. gracilicarpus* primarily by the relative short size of the carpus of the first pereiopods (about 1.1 times the chela length in *M. kimwoni* **sp. nov.**, vs. about 1.4 in *M. gracilicarpus*), as well as the proportions of the carpus of the minor second pereiopod (0.8 times chela length in *M. kimwoni* **sp. nov.** vs. about 1.5 in *M. gracilicarpus*). The new species can be distinguished from *M. brucei* by the presence of a minute tubercle on the chela of the major second pereiopod (vs. absent in *M. brucei*), relatively long fingers of first pereiopod (about 0.6 of palm length in *M. kimwoni* **sp. nov.** vs. about 0.5 in *M. brucei*), relatively short carpus of major second pereiopod (about 0.4 of the palm length in *M. kimwoni* **sp. nov.** vs 0.6 in *M. brucei*) and the relatively long carpus of the minor second

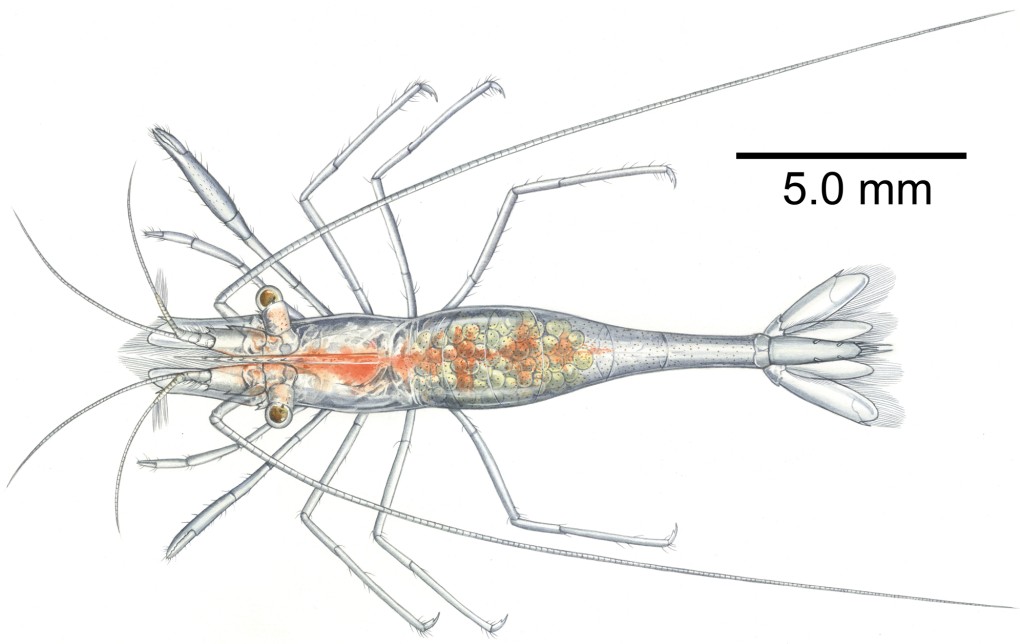

**Figure 13 Artistic interpretation of *Mesopontonia kimwoni* sp. nov.** Holotype, ovigerous female, pocl 2.5 mm (NIBRIV0000862985). Painting by Yun Kyoung Kim.

pereiopod (about 1.3 times the palm length in *M. kimwoni* **sp. nov.** vs. about 0.9 in *M. brucei*).

*M. kimwoni* **sp. nov.**, appears most closely related to the west Indian species, *M. brevicarpus*, sharing a similar rostral formulation, a tuberculate major second pereiopod chela, as well as the ratio of the ambulatory pereiopods. Both species can be most easily distinguished on the basis of the combination of the following characters, (1) fingers of first chela with entire cutting edge (vs. fine pectinated serrations subapically on both fingers in *M. brevicarpus*); (2) hepatic tooth reaching to the anterior margin of the carapace (vs. reaching or extending to anterior margin of carapace in *M. brevicarpus*) and (3) the finger of minor chela being about 0.7 of the palm length (vs. fingers subequal to palm in *M. brevicarpus*).

*Bruce (1996)* suggested two further new species may be present in the genus, one collected from Indonesia (*Bruce, 1996*), as well as the juvenile specimen assigned to *M. gorgoniophila* in *Bruce (1985)*, both however were left unnamed. Given their incomplete or juvenile status these taxa are not considered herein, but are unlikely to be the same species as *M. kimwoni*, as the details of the first and second pereiopods are different.

### Molecular data analyses

Fragments of 658 and 462 bp were obtained for the COI and 16S markers, respectively. The multiple sequence alignment revealed that the K2P distance between the five specimens of *M. verrucimanus* which showed minor morphological variations in the dentition of the rostrum and proportions of the second pereiopods fall within an intraspecific range, being

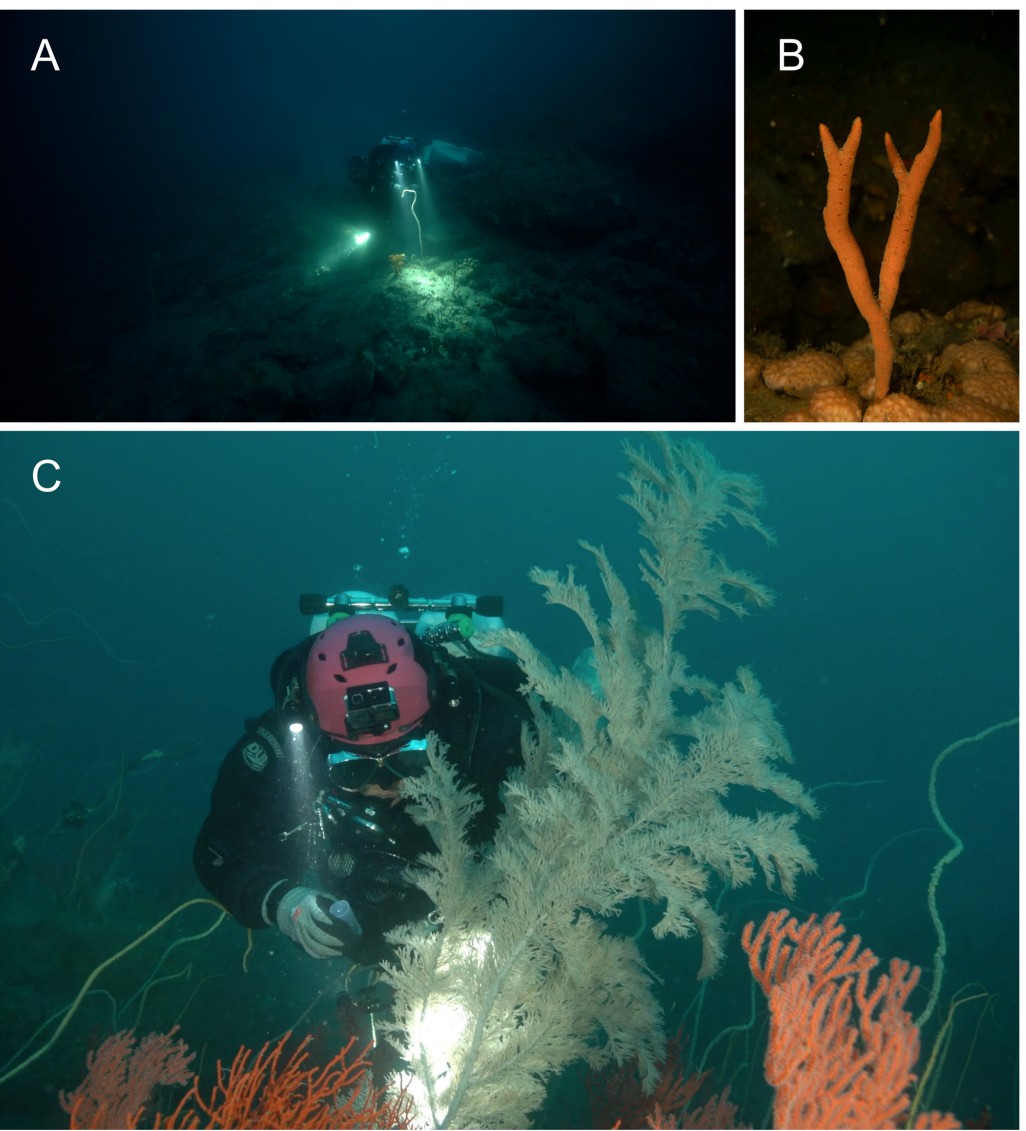

**Figure 14 Habitat and host of *Mesopontonia* from Munseom Islet, Jejudo Island, Korea.** Habitat and host specimens of *M. verrucimanus* (A, B) and *M. kimwoni* **sp. nov.** (C). (A) host sea whip, *Ellisella* cf. *limbaughi* and habitat in depth of 57 m; (B) host sponge *Raspailia (Raspaxilla) hirsuta* and habitat in depth of 55 m; (C) host black coral *Myriopathes lata* and habitat in depth of 55 m. Photographic Credits: Jong Moon Choi.

0–0.5% (Table 2). The intraspecific divergence between both specimens of *M. kimwoni* **sp. nov.** was higher at 1.1% (Table 2).

To eludicate the phylogenetic position of the genus, an analysis was performed on 22 specimens of 16 species of 12 genera (Table 1). The ML and BI analyses showed the same topology (Fig. 15), and the combined phylogenetic tree clearly demonstrated the monophyly of *Mesopontonia* with high support values (BP = 100, PP = 100). Furthermore, their distant relationship was supported by the K2P distance, which was 13.6% (Table 2).

**Table 2 Pairwise distances of COI sequences and selected morphological characteristics for specimens used in the analysis (P2 - second pereiopods).**

| | Species | Rostral formula | Shape and size of P2 | 1 | 2 | 3 | 4 | 5 | 6 |
|---|---|---|---|---|---|---|---|---|---|
| 1 | *M. kimwoni* **sp. nov.** (1) | 1 − 8/2 | Unequal and dissimilar | | | | | | |
| 2 | *M. kimwoni* **sp. nov.** (2) | 1 − 8/2 | Unequal and dissimilar | 0.011 | | | | | |
| 3 | *M. verrucimanus* (1) | 1 − 8/1 | Unequal and dissimilar | 0.142 | 0.1360 | | | | |
| 4 | *M. verrucimanus* (2) | 1 − 8/0 | Equal and similar | 0.1380 | 0.1320 | 0.0030 | | | |
| 5 | *M. verrucimanus* (3) | 1 − 9/0 | Unequal and dissimilar | 0.1380 | 0.1320 | 0.0030 | 0.0000 | | |
| 6 | *M. verrucimanus* (4) | 2 − 8/1 | Unequal and dissimilar | 0.1400 | 0.1340 | 0.0050 | 0.0020 | 0.0020 | |
| 7 | *M. verrucimanus* (5) | 1 − 9/2 | Equal and similar | 0.1380 | 0.1320 | 0.0030 | 0.0000 | 0.0000 | 0.0020 |

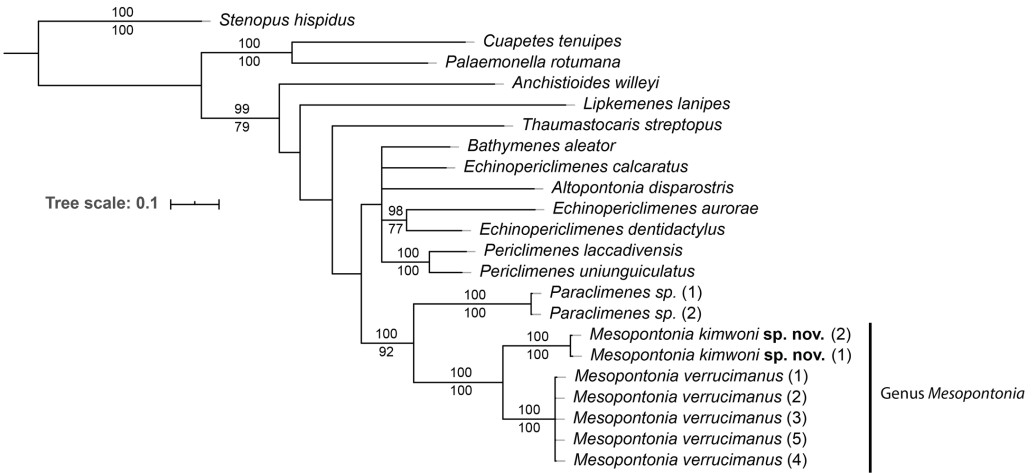

**Figure 15 Phylogenetic tree obtained by the Maximum likelihood (ML) analysis based on the combined dataset for COI and 16S sequences.** Numbers at nodes represent Maximum Likelihood bootstrap percentage (above) and Bayesian posterior probabilities (below), numbers less than 75% are not shown.

From the concatenated tree in the present analysis, the genera *Mesopontonia* and *Paraclimenes* are postulated to be sister taxa with high support values (BP = 100, PP = 92), indicating that they are more genetically related to each other than the remaining analysed genera, supported by morphological similarities.

## DISCUSSION

The present study explored the commensal palaemonid fauna of Jejudo Island, recording *Mesopontonia kimwoni* **sp. nov.** and *M. verrucimanus* at higher latitude temperate waters than the genus was previously known from. While the six species previously known in the genus had been reported from between 117 and 600 m depth by trawling and dredging (*Bruce, 1967*; *Bruce, 1979*; *Bruce, 1984*; *Bruce, 1985*; *Bruce, 1990*; *Bruce, 1991*; *Bruce, 1996*; *Burukovsky, 1991*; *Li & Bruce, 2006*), the present specimens were collected from shallower depths of less than 75 m. As they were directly collected with technical SCUBA diving

equipment, more details are available on their habitat and ecology, whilst color patterns are recorded for the first time for the genus as a whole.

*Mesopontonia kimwoni* **sp. nov.** can be distinguished from all other *Mesopontonia* species by the combination of the biunguiculate dactylus of the ambulatory pereiopods, the lack of a dorsolateral dactylar flange on the major second chela, the relatively long carpus of the minor second pereiopod and the entire cutting edge of fingers of first chela. Specimens of *M. verrucimanus* exhibited minor morphological variation in rostral dentition and proportions of the major second pereiopod, but all specimens are clearly conspecific.

Due to its rarity, *Mesopontonia* had not been previously included in family level phylogenies (e.g., *Kou et al., 2013*; *Gan et al., 2015*; *Horká et al., 2016*), but is herein shown to be phylogenetically close to *Paraclimenes*. This is also supported by a relatively similar morphology with both antennal and supraorbital teeth being absent; and the epigastric and hepatic teeth being present. Nevertheless, *Paraclimenes* can be readily distinguished from *Mesopontonia* by the presence of a well-developed exopod on the third maxilliped (*Bruce, 1995*).

**Key to species of *Mesopontonia* Bruce, 1967 (Adapted from *Bruce, 1996*; *Li & Bruce, 2006*)**

| | | |
|---|---|---|
| 1. | Dactylus of ambulatory pereiopods simple | 2 |
| – | Dactylus of ambulatory pereiopods biunguiculate | 3 |
| 2. | Major second pereiopod with dorsolateral dactylar flange ................................ *M. monodactylus* Bruce, 1991 | |
| – | Major second pereiopod without dorsolateral dactylar flange ................................ *M. verrucimanus* Bruce, 1996 | |
| 3. | Major second pereiopod with dorsolateral dactylar flange ................................ *M. gorgoniophila* Bruce, 1967 | |
| – | Major second pereiopod without dorsolateral dactylar flange | 4 |
| 4. | Carpus of minor second pereiopod longer than chela ................................ *M. gracilicarpus* Bruce, 1990 | |
| – | Carpus of minor second pereiopod shorter than chela | 5 |
| 5. | Carpus of minor second pereiopod shorter than palm ................................ *M. brucei* Burukovsky, 1991 | |
| – | Carpus of minor second pereiopod longer than palm | 6 |
| 6. | Fingers of first chela with pectinated serrations, subapically ................................ *M. brevicarpus* Li & Bruce, 2006 | |
| – | Fingers of first chela with clear entire cutting edge ................................ *M. kimwoni* **sp. nov.** | |

**Abbreviations used in text, tables and figures**

| | |
|---|---|
| **BI** | Bayesian Inference |
| **BP** | Maximum likelihood bootstrap percentage |
| **K2P** | Kimura 2-parameter |
| **ML** | Maximum likelihood |

| POCL | postorbital carapace length |
|------|------------------------------|
| PP | Bayesian posterior probabilities |
| R | Rostrum formula; formulation of epigastric tooth and teeth on rostrum |

## ACKNOWLEDGEMENTS

We thank Dr. Hyi-Seung Lee (KISOT, Korea) and Dr. Wilfredo L. Campos (UPV, Philippines) for their invitation and management in Philippines fieldwork. We are also grateful to Dr. Benny K.K. Chan (Academia Sinica, Taiwan) for the invitation to participate in the exploration of Green Island, Taiwan. Authors also wish to thank to Prof. Sung-Jin Hwang (Woosuk University, Korea) and Dr. Hyung June Kim (MABIK, Korea) for helping identification of the host species, and Dr. Sanghui Lee (MABIK, Korea) and Dr. Damin Lee (SNU, KOREA) for their assistance in field survey, and to Ms. Eunseon Jun (Sahmyook University, Korea) for helping with molecular experiments. TP would like to also thank Ms. Yun Kyoung Kim for her worthful assistance with the color drawing in Fig. 13. We extend sincere thanks to Capt. Jong Moon Choi (Blue Research, Korea) for his kind help of photographing the habitats and guidance in Deep-Sea Research Diving.

### Funding

This work was supported by grants of the National Institute of Biological Resources (NIBR201902107 and NIBR202010102), and the Marine Biotechnology Program of Marine Biotechnology Program (No. 20170431) of the Korean Government. The funders had no role in study design, data collection and analysis, decision to publish, or preparation of the manuscript.

### Grant Disclosures

The following grant information was disclosed by the authors:
The National Institute of Biological Resources: NIBR201902107, NIBR202010102.
The Marine Biotechnology Program of (No. 20170431) of Korean Government.

### Competing Interests

The authors declare there are no competing interests.

### Author Contributions

- Jin-Ho Park conceived and designed the experiments, performed the experiments, analyzed the data, prepared figures and/or tables, authored or reviewed drafts of the paper, investigation, and approved the final draft.
- Sammy De Grave analyzed the data, authored or reviewed drafts of the paper, and approved the final draft.
- Taeseo Park analyzed the data, authored or reviewed drafts of the paper, funding acquisition, Investigation, and approved the final draft.

## Field Study Permissions

The following information was supplied relating to field study approvals (i.e., approving body and any reference numbers):

Field experiments were approved under Korea (National Institute of Biological Resources, Jeju Special Self-Governing Province), Palau (Bureau of Marine Resources, Ministry of Resources) and Philippines (Development and Bureau of Fisheries and Aquatic Resources, Department of Agriculture), respectively. Permission to collection was granted by Korea (No. 577, 2019 and No. 13895, 2019), Palau (No. Re-19-26, 2019), Philippines (No. 07-BOHFO-111/19, 2019).

## Data Availability

The raw measurements are available in the Supplemental Files. The sequences are available at GenBank: MT311862–MT311873 and MT311971–MT311981.

## New Species Registration

The following information was supplied regarding the registration of a newly described species:

Publication LSID: urn:lsid:zoobank.org:pub:3CB43670-472F-49AE-80F2-EAE9597E12BD

*Mesopontonia kimwoni* sp. nov LSID: urn:lsid:zoobank.org:act:BBA317A3-7140-4D97-BCF4-DB6EEF6617F5.

## Supplemental Information

Supplemental information for this article can be found online at http://dx.doi.org/10.7717/peerj.10190#supplemental-information.

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
