# Peer review of "On the genus Mesopontonia Bruce, 1967 (Crustacea: Decapoda: Palaemonidae) in Korea, with the description of a new species"

_PeerJ, doi:10.7717/peerj.10190_

## Round 0.1 · original submission · Major Revisions

Thank you very much for the interesting manuscript that you had a description of a new species. However, the present form of this manuscript is not ready to publish in PeerJ. Concluded from our reviewers, you have to revise a manuscript according to all comments and resubmit a revision version as soon as possible. Please, address all comments from the reviewers because it will fulfill the scientific backgrounds of your manuscript. I am waiting for a revised version.

Reviewer 1 ·

Basic reporting

The article should be re-evaluated for professional English usage. Sentences are long and complex, making understanding of the context difficult. The unit of each quantity should always be informed.

Experimental design

No comment

Validity of the findings

No comment.

Additional comments

Morphological and genetic results are convining to prove that the creature is a new species.
However, the language usage is still needed to be re-checked for complex and non-informative sentences.

Reviewer 2 ·

Basic reporting

This manuscript describes a new species and a new record of Mesopontonia shrimps in Jeju island, Korea. The manuscript is largely taxonomic based, contains only one new species and new records. Readers of Peer J include colleagues from wide ranges of fields. One of the attractive point of this article is these shrimps is from mesophotic coral reefs in Korea, which is rather unknown habitats and with understudied diversity. I think if it is to be accepted in Peer J, it should be value-added by mentioning these species are from mesophotic coral reefs, mesophotic zone coral reef ecology in Jeju and with some more discussion on the reproductive period of Mesopontonia verrucimanus in Jeju at the end of discussion.

Experimental design

The paper is taxonomic based, no comments on experimental design.

Validity of the findings

see my comments for author below

Additional comments

This manuscript describes a new species and a new record of Mesopontonia shrimps in Jeju island, Korea. The manuscript is largely taxonomic based, contains only one new species and new records. Readers of Peer J include colleagues from wide ranges of fields. One of the attractive point of this article is these shrimps is from mesophotic coral reefs in Korea, which is rather unknown habitats and with understudied diversity. I think if it is to be accepted in Peer J, it should be value-added by mentioning these species are from mesophotic coral reefs, mesophotic zone coral reef ecology in Jeju and with some more discussion on the reproductive period of Mesopontonia verrucimanus in Jeju at the end of discussion. The MS will need revisions according to my comments below. I am happy to read the revised manuscript again.

Specific comments are:

1) The name “Jejudo Island” – The authors can consider to use the name Jeju Island. Jeju Island appears to be more commonly found in the internet and scientific literatures. The word ‘do’ in Jejudo already carry the meaning of ‘Island’ in Korean language, so it is appropriate to name it as Jeju Island.

2) Lines 70-72, the authors stated “The sea area around the island is largely influenced by the warm, saline Tsushima Warm Current, a branch of the Kuroshio Current, flowing northeastward through the Korea Strait from the East China Sea (Lie et al. 2000).” The authors can modify this sentence. In summer, Tsushima current is mixed with low-salinity, high-turbidity waters from the Yangtze River to influence the waters in Jeju Island. In winter, the Yangtze River discharge reduced and result in higher salinity (Rebstock and Kang, 2003; Lim et al. 2019). Lim et al. (2019) which contains plots of salinity in summer and winter and also surface temperature in Munseom. Also see Rebstock and Kang, 2003 for currents (hyperlinks for literatures attached below for your references). These references should be cited in text.

3) Lines 72-74, the authors stated “Munseom Islet (Fig. 1C) is located off the south coast of the main island, and consists of volcanic rocks covered with rich invertebrate communities” I would suggest citing three references as “Munseom Islet (Fig. 1C) is located off the south coast of the main island, and consists of volcanic rocks covered with rich invertebrate communities (Cho et al. 2014, Lutaenko et al. 2019; Lee et al. 2019).” Cho et al. (2014) contains a survey of diversity of invertebrates in Munseom. Lutaenko reported at least five new species of bivalves in Jeju. Even a new species of brittle star is recently identified from Munseom suggesting its high invertebrate diversity here. (hyperlinks for literatures attached below for your references)

4) Starting from line 93, I would suggest the authors use a few sentences to describe a general diversity of corals, sponges and soft coral species in the mesophotic environment they dived in Jeju. Diversity and ecology of Mesophotic environment received very little attention. Even the book “Mesophotic Coral Ecosystems” (https://www.springer.com/gp/book/9783319927343) contains chapters of mesophotic coral communities in Okinawa, Japan and Taiwan but without Korea. So, having some information about the mesophotic zone and with some photos of the habitat if possible can enhance this article not solely interested by specialist taxonomist.

5) Figure 9 – 11 illustrate the morphological part of the new species. I would suggest in these figures, use arrows to indicate diagnostic part which are unique for this species. In the remarks of M. kimwoni sp. nov, the authors stated how this species is different from other described closely related species. Since this is a submission to Peer J, readers of Peer J is not specialist in specific taxonomic group. I would suggest the authors to make an additional figure, with some diagrammatic line drawings for other species and show how the new species can be distinguished from others. So, non-specialists can also follow the identifications.

6) Lines 144-185, I can see authors has described whether females are ovigerous or not from the specimens of Mesopontonia verrucimanus collected in Munseom from various months in 2018 and 2019. I would suggest to plot a simple horizontal bar chart covering the year 2018 and 2019, blacken the bars in those months when the females are ovigerous. This can show the occurrence of ovigerous females in summer months but not winter months. This showed that the reproductive cycle of shrimps in mesophotic zone are seasonal. Comparisons can also be made to subtidal bivalves in Munseom (Fig. 10 in Lim et al. 2019). Subtidal communities in Jeju have mature gonads in June to August. This period overlapps a portion with period of ovigerous Mesopontonia verrucimanus, but M. verrucimanus extend its reproductive season to September, slightly longer than subtidal bivalves. An additional paragraph in the discussion can be made to discuss the reproductive cycle of females of M. verricimanus and compared the pattern with subtidal bivalves in Jeju.

References

Cho et al. 2014 https://www.researchgate.net/publication/262880572_A_study_on_the_biodiversity_of_benthic_invertebrates_in_the_waters_of_Seogwipo_Jeju_Island_Korea

Lutaenko et al. 2019
https://www.researchgate.net/publication/334230937_Lutaenko_KA_Noseworthy_RG_Choi_K-S_Marine_bivalve_mollusks_of_Jeju_Island_Korea_Part_1_The_Korean_Journal_of_Malacology_2019_v_35_N_2_pp_149-238

Lim et al. 2019
http://zoolstud.sinica.edu.tw/Journals/58/58-29.pdf

Lee et al. 2019
http://zoolstud.sinica.edu.tw/Journals/58/58-08.pdf

Rebstock and Kang, 2003
https://www.sciencedirect.com/science/article/abs/pii/S0079661103001769

Reviewer 3 ·

Basic reporting

English in this manuscript should be edited by the expert in order not to make any confusion in the sentences.
Literature references, figures, and tables are suitable for publication.

Experimental design

Mitogenome sequencing technique is well established in numerous publications and they used the typical methods. Therefore, there is nothing to commend about experimental design.

Validity of the findings

Reporting two additional mitogenome sequences in the Dendrobranchiata is the main topic in this manuscript. They found two mitogenome sequences are the same as the previous studies in the overall structures, gene organization, and phylogenetic relationship. therefore there is few novel findings in the manuscript except for mitogenome sequences. If the policy of this manuscript is OK, overall writing and structures for this manuscript are typically acceptable.

Additional comments

Reporting two additional mitogenome sequences in the Dendrobranchiata is the main topic in this manuscript. The authors found two mitogenome sequences are the same as the previous studies in the overall structures, gene organization, and phylogenetic relationship. Therefore there are few novel findings in the manuscript except for adding mitogenome sequences. If the policy in this manuscript accepts this kind of manuscript, overall writing and structures for this manuscript are typically acceptable. Their contribution is the addition of two mitogenome sequences in the taxa.

Reviewer 4 ·

Basic reporting

no comment

Experimental design

no comment

Validity of the findings

Representatives of the deep-water pontoniine shrimp genus Mespontonia are quite rare for scientists and any records of these shrimps are extremely interesting. However, this is also a problem for describing new species. All previously described species are extremely morphoogically close, while, that is more important, intraspecific variability of the morphology are absent for most species. In this regard, it is difficult to determine any stable and important diagnostic morphological features within the genus.
This presented article describes well describe the found shrimpss and the morphological differences between studied species. In addition, genetic differences are shown.
At the same time, the genus Mesopontonia currently includes 6 valid species, which, like most caridean species in the Indo-West Pacific, can be found in any part of this zoogeographical region. Morphological differences from other species in the article are described and discussed very poorly and only in general features (like length\width ratios nd measurements). Moreover, some species (for example, M. brevicarpus, which seems to be most morphologically close to the newly described species) were initially poorly described (see Li, 2006), and many morphological features have not been specified ince the original description. Most of species within the genus need careful morphological re-description. Also, the presented genetic data and the absence of such data for other species of the genus do not allow to conclude that authors are dealing with different biological species.
In this regard, I cannot unfortunately conclude that the described species is really new to science. For me, the morphological features of the species from other species of the genus, for example, M. brevicarpus, are not obvious, and not confirmed in any way. The discussed morphological structures can be found in large/small individuals and no previous studies have indicated them as diagnostic and sufficient to identify the new species.
In the case of genus Mesopontonia, the description a new species need the complete revision of the genus (better, using molecular data), which is very likely to lead its significant reduction. However, at least a revision of the morphological features of all the described 6 species is extremely important for the study of the diversity of the genus.

Final decision - the article cannot be published with a description of the new species (as presented), since its separation is not confirmed by either morphological or genetic data.

Additional comments

Representatives of the deep-water pontoniine shrimp genus Mespontonia are quite rare for scientists and any records of these shrimps are extremely interesting. However, this is also a problem for describing new species. All previously described species are extremely morphoogically close, while, that is more important, intraspecific variability of the morphology are absent for most species. In this regard, it is difficult to determine any stable and important diagnostic morphological features within the genus.
This presented article describes well describe the found shrimpss and the morphological differences between studied species. In addition, genetic differences are shown.
At the same time, the genus Mesopontonia currently includes 6 valid species, which, like most caridean species in the Indo-West Pacific, can be found in any part of this zoogeographical region. Morphological differences from other species in the article are described and discussed very poorly and only in general features (like length\width ratios nd measurements). Moreover, some species (for example, M. brevicarpus, which seems to be most morphologically close to the newly described species) were initially poorly described (see Li, 2006), and many morphological features have not been specified ince the original description. Most of species within the genus need careful morphological re-description. Also, the presented genetic data and the absence of such data for other species of the genus do not allow to conclude that authors are dealing with different biological species.
In this regard, I cannot unfortunately conclude that the described species is really new to science. For me, the morphological features of the species from other species of the genus, for example, M. brevicarpus, are not obvious, and not confirmed in any way. The discussed morphological structures can be found in large/small individuals and no previous studies have indicated them as diagnostic and sufficient to identify the new species.
In the case of genus Mesopontonia, the description a new species need the complete revision of the genus (better, using molecular data), which is very likely to lead its significant reduction. However, at least a revision of the morphological features of all the described 6 species is extremely important for the study of the diversity of the genus.

Final decision - the article cannot be published with a description of the new species (as presented), since its separation is not confirmed by either morphological or genetic data.

---

## Round 0.2 · Minor Revisions

Thank you very much for your significantly improve this manuscript. However, I would like you to improve your figures; in figures 2, 3, 4, 5, 6, 7, 8, 9, 10, 11, and 13. I need you to point the importance structure for the reader, better than figure without the label.
I am looking for words to see your revision version.

Reviewer 2 ·

Basic reporting

The authors has addressed major part of the comments and the MS can be accepted. Of course I agree the illustration is at the professional level. Whether the authors is encouraged to add in more ecological and illustration for non-specialists/beginner to understand is really depends the editor's decision on the scope of PeerJ - a really specialist taxonomy journal or a biodiversity, taxonomy and ecology journal for both specialists and general readers.

Experimental design

Addressed

Validity of the findings

Addressed

Additional comments

The authors has addressed major part of the comments and the MS can be accepted. Of course I agree the illustration is at the professional level. Whether the authors is encouraged to add in more ecological and illustration for non-specialists/beginner to understand is really depends the editor's decision on the scope of PeerJ - a really specialist taxonomy journal or a biodiversity, taxonomy and ecology journal for both specialists and general readers.

---

## Round 0.3 · accepted · Accept

Thank you very much for your response. Congratulations to you and your team.